# Debiasing Machine Learning Models by Using Weakly Supervised Learning

## Abstract

We tackle the problem of bias mitigation of algorithmic decisions in a setting where both the output of the algorithm and the sensitive variable are continuous. Most of prior work deals with discrete sensitive variables, meaning that the biases are measured for subgroups of persons defined by a label, leaving out important algorithmic bias cases, where the sensitive variable is continuous. Typical examples are unfair decisions made with respect to the age or the financial status. In our work, we then propose a bias mitigation strategy for continuous sensitive variables, based on the notion of endogeneity which comes from the field of econometrics. In addition to solve this new problem, our bias mitigation strategy is a weakly supervised learning method which requires that a small portion of the data can be measured in a fair manner. It is model agnostic, in the sense that it does not make any hypothesis on the prediction model. It also makes use of a reasonably large amount of input observations and their corresponding predictions. Only a small fraction of the true output predictions should be known. This therefore limits the need for expert interventions. Results obtained on synthetic data show the effectiveness of our approach for examples as close as possible to real-life applications in econometrics.

## 1 Introduction

Machine Learning (ML) provides a way to learn accurate statistical models able to learn tasks, such as classification, regression, forecasting, recommendation etc., from data. Despite the flexibility of ML models and their high accuracy, such algorithms also present some drawbacks. One of the mostly discussed subject over the past few years is how ML models can produce biased social decisions, *i.e.* outcomes that systematically favors some groups to the detriment of others (rich and poor people, for example), based on some variables referred to as sensitive attributes, which should not play any role in the decision. Such biases lead to ethical concerns, which have turned to be legal concerns for critical applications due to the new regulations on the use of Artificial Intelligence. A typical example is the A.I. act[1], which will expose to severe sanctions the companies selling unreasonably biased AI systems in the European-Union, if these systems are used to high-risk applications. Initiated in Dwork et al. (2012), we then refer for instance to Oneto & Chiappa (2020), Barocas et al. (2023; 2017), Besse et al. (2022), Bird et al. (2020), Mehrabi et al. (2021) or Del Barrio et al. (2020) and references therein for important strategies dedicated to mitigate algorithmic biases. If an A.I. system turns out to be biased, such strategies are indeed of primary importance to make the systems compliant with the regulations. Let us develop the presentation of these strategies. When a notion of bias in the algorithm has been defined and chosen, there are a variety of techniques to mitigate model bias, which can be split into three main categories Mehrabi et al. (2021):

1. Pre-processing techniques: since the data can be biased, for historical reasons, misrepresentation, or more intricate patterns, the use of such data can render the model unfair. Therefore, treating the data before it feeds the model is a possible strategy to mitigate the bias in the decisions Kamiran & Calders (2012); Feldman et al. (2015a); Calmon et al. (2017); Samadi et al. (2018); Gordaliza et al. (2019).

---

[1] https://eur-lex.europa.eu/legal-content/EN/TXT/?uri=CELEX:52021PC0206 (accessed on 24 July 2023)

2. In-processing techniques: to reduce the biased decisions, these techniques aim at changing the model training procedure, by adapting the objective function, by adding constraints, or by doing both Calders & Verwer (2010); Kamishima et al. (2012); Zafar et al. (2017); Risser et al. (2022b).

3. Post-processing techniques: when one has a black-box model that cannot be changed, the only way to possibly reduce the bias is by using post-processing techniques. In this case, the outputs produced by the model are processed once again, to be less biased Kamiran et al. (2010); Hardt et al. (2016); Woodworth et al. (2017).

In the vast majority of the cases, the problem of Fair Machine Learning deals with classification tasks, with discrete sensitive attributes (such as gender, or race). In this scenario, the model's outputs take values on a discrete set (representing the possible outcomes) and the same occurs for the sensitive attribute ($S = 0$ representing the protected group, $S = 1$ representing the privileged one). The usual measures of bias introduced in this setting consist in evaluating the proportion of individuals belonging to each sub-group ($S = 0$ or $S = 1$) assigned to each class. In this framework we can use measures such as Disparate Impact Feldman et al. (2015b) — which compares the proportion of individuals who receive a positive outcome in the privileged and to the ones assigned to the same outcome but belonging to the unprivileged group — Equalized Odds Hardt et al. (2016) — which compares the true positive and false positive rates between the two groups —, Equal Opportunity Hardt et al. (2016) — a measure that aims at matching the true positive rates for different values of the protected attribute — or Treatment Equality Berk et al. (2021) — which measures the difference between the ratio of false negatives to false positives between both groups.

However, in forecasting applications, where the objective is to produce a score that suitably summarizes the input data, the model's output referred to as $Y(x)$ no longer takes values on a discrete set but in a continuous interval. Hence, $Y(x) \in [a, b]$, where $x$ represents all attributes (sensitive and non-sensitive ones). This renders the evaluation of the model's fair behavior more difficult, since it is hard to ensure that the values of $Y(x)$ are the same for $x$ belonging to different sub-groups. In this scenario, measures like Fairness through Awareness Dwork et al. (2012) and Counterfactual Fairness Kusner et al. (2017) are interesting options, since their strategy to mitigate bias does not rely on some discretization of the attributes, but instead they are able to deal with continuous-valued sensitive attributes. Note also that Oneto et al. (2020) can also handle the case of continuous sensitive variables by using a discretization strategy.

The problem becomes even more challenging when we model the sensitive attributes no longer as a discrete variables but as continuous ones. This is a suitable choice to model sources of bias encoded in characteristics as age, financial status or ethnic proportions. In such a case, the aforementioned measures of fairness cannot be applied, since it is not possible to separate the population into sub-groups and assess the model's performance on each of them.

In such a setting, *i.e.*, forecasting with continuous-valued sensitive attributes, the situation can be modeled as a regression problem where there is correlation between the independent variables and the error, establishing a dependency between them, a phenomenon known as endogeneity. For further discussions and analysis about this phenomenon in econometrics, we refer to Nakamura & Nakamura (1998) or Florens (2003).

The objective of this work is to mitigate bias of forecasting models, when dealing with continuous-valued sensitive attributes and continuous-valued predictions. Here we focus on the case where we need to mitigate the bias of an already operating model, that should be treated as a black box. Since we can neither change the model's input nor its training procedure, we propose here a post-processing treatment.

Because it is not possible to separate the population in sub-groups, in order to evaluate if the model is biased or not, we need an external source of knowledge to evaluate whether the produced outcomes are fair or not (*i.e.*, to assess if the model is systematically treating differently some groups of the population). In this work, we encode this external knowledge by two means. First, we assume that a group of specialists (composed of economists, sociologists, lawyers, and others) provides us with the probability distribution of scores that a fair model should follow. Second, we also have access to an oracle/specialist that given a particular person returns the fair score for such an individual, but we can use such a specialist only for a few individuals. By using such an approach, we cope with the problem of bias mitigation in two steps: first, we need to know the unbiased scores; second, we need to properly distribute those scores among the individuals of the population.

The same ideas are developed in bias mitigation for rankings of recommender systems Wang et al. (2023). In this case, as pointed out also in Kletti et al. (2022), there exists a prior of what should be a fair ranking. Enhancing fairness is thus achieved by comparison between this ideal fair scores and the observations.

To better contextualize the applicability of our framework, let us consider the problem of risk assignment made by assurance companies. Note that this application may have a strong impact on individuals' lives and will therefore be likely to be ranked as High risk by the A.I. act, so they will be regulated by the articles 9.7, 10.2, 10.3 and 71.3 of this act. In the risk assignment case, we know the distribution of the risk scores for a particular city, and we know that it is biased. This could be the case for a smaller city, for which we observe more frequently than for big cities the occurrence of high values of risk scores. With our framework, we compare the distribution of the risk scores of this small city with its "ideal version", *i.e.*, we compare such a distribution with the one that should have been observed if the living place was not, or if it was fairly, taken into account. Besides the comparison with the "ideal version" of the population, we also use information obtained from specialists to know the fair risk scores for some individuals. This procedure is equivalent to a polling, where a group of interviewers (recruited by the assurance company or for a group of auditors) collects the relevant information (such as profession, age, driving history, among others) about a little fraction of the population (since it is an expensive and time-consuming procedure), and then assigns to them the fair risk scores.

**Layout of the Paper**

The paper is organized as follows: in Section 2, we model the problem of mitigating bias in a black box model, formalizing the idea of endogeneity and how it is usually treated in economics. In Section 3 we present our methodology to automatically mitigate the bias, inspired by recent results in Inverse Problems; we also present a theoretical analysis of our approach, assessing its performance. In Section 4 we evaluate our approach by means of numerical simulations, considering 1- and 2-dimensional signals, representing the cases where we have a single sensitive attribute or two of them, respectively. Finally, in Section 5 we present the conclusions of this work and perspectives for future ones.

## 2 Theoretical Background

As presented in Section 1, ML models can produce decisions that may convey biased information, learned from many different sources of bias encountered at the different stages of the data processing. Our focus here is to mitigate the bias of an algorithm (*i.e.* the automated systematically discriminatory treatment among population subgroups) by post-processing its outputs. To better understand it, let us consider an ML model, which acts a black box model where an observation $\mathbf{x}$ is transformed into a continuous value $\mathbf{x} \mapsto Y(\mathbf{x})$.

Such a model, that will perform a forecasting task Bishop & Nasrabadi (2006), takes a set of characteristics (such as financial status, gender, age, education level, country, etc.), represented by $\mathbf{x} \in \mathbb{R}^P$ and performs a statistical treatment on such data. Since here we are treating the mitigation of bias of supervised ML models, we do not know, explicitly, how the ML model assigned the observed scores. In fact, we only know that it consists in an automated procedure, based on a flexible enough parameterized model, whose parameters were optimized in order to satisfy a specific criterion, typically using supervised learning Goodfellow et al. (2016). After such a procedure, the model outputs a score $Y(\mathbf{x}) \in \mathbb{R}$, summarizing the collected information in a suitable way to take a decision, for credit assignment or selection of students to universities, for example.

Since the model was trained automatically, usually in a very high dimensional space, it could have learned, in an unwanted way, how to satisfy the training criterion by favouring a group of individuals to the detriment of another. We refer for instance to Bell & Sagun (2023) or Risser et al. (2022a) for the description of such an optimization drawback. When it happens, the characteristic that drives an unwanted change of behaviour and is at the origin of the algorithmic bias, is called the sensitive attribute. Removing this effect and thus mitigating the corresponding bias has become a legal constraint when the sensitive variable is a prohibited variable. Many sensitive variables are discrete variables such as gender, religious orientation or race, but continuous-valued sensitive variable may also be taken into account such as age, recidivism score in predictive police, rate of inhabitants with some particular ethnic origin or rate of unemployment of a some districts that serve as proxy for social or ethnic origin for instance.

Because here we deal with the problem of an already operating model, we can neither change its input, $\mathbf{x}$, by transforming it in a suitable manner in order to reduce the impact of the sensitive attributes, nor change the way the model was trained, by changing the training criterion, by adding constraints or by doing both. In this scenario, we must treat the model as a black box and only treat its biased output, $Y(\mathbf{x})$.

Here we model the biased output as the sum of two terms

$$Y(\mathbf{x}) = \varphi^*(\mathbf{x}) + U.$$

Note that we can interpret this model as follows. The term $\varphi^*(\mathbf{x})$ represents the output of the algorithm that should have been obtained by the model, if it was not biased at all, and $U$ is a type of measurement noise, that may affect differently the different groups of the population, *i.e.*, it reflects a dependence with respect to the input attributes,

$$\mathbb{E}[U|\mathbf{x}] \neq 0,$$

which implies that $U = U(\mathbf{x})$, leading to biased models

$$\underbrace{Y(\mathbf{x})}_{\text{Observed Biased Score}} = \underbrace{\varphi^*(\mathbf{x})}_{\text{Unbiased Score}} + \underbrace{U(\mathbf{x})}_{\text{Bias Term}}. \tag{1}$$

Since the property $\mathbb{E}[U|X] = 0$ is not verified, $\varphi^*(\mathbf{x})$ is not the conditional expectation of $Y$ given $\mathbf{x}$. For example, the noise $U$ may depend on some characteristic of the individual which is unobservable for the statistician, but known from assignment priors of the treatment. The choice of the levels $\mathbf{x}$ than depends on this characteristic, and then process a dependence between $U$ and $\mathbf{x}$. Note that in this work, the model (1), we tackle consider the case of continuous-valued sensitive attributes, which are particularly useful to model financial status, age or ethnic proportions Mary et al. (2019). Bias with respect to continuous sensitive attribute has received scarce attention in the fairness literature where bias is often conveyed by a discrete variable that splits the population into subsamples. Also, neither $\varphi^*(\mathbf{x})$ nor $U(\mathbf{x})$ are directly observed. The goal, therefore, is to reduce as much as possible the effect of $U(\mathbf{x})$ on $Y(\mathbf{x})$, which models that bias, and by doing so, to estimate $\varphi^*(\mathbf{x})$ .

Since $U(\mathbf{x})$ is a measurement noise, but correlated with $\mathbf{x}$, we model it as

$$U(\mathbf{x}) \sim N(\mu(\mathbf{x}), \sigma(\mathbf{x})) \tag{2}$$

such that the bias term follows a Gaussian distribution, whose mean $\mu(\mathbf{x})$ and variance $\sigma(\mathbf{x})$ encode its dependence on $\mathbf{x}$.

Solving the estimation problem directly from the observations $Y(\mathbf{x}_i)$, $i = 1, \ldots, n$ leads to a solution which is not the unbiased response $\varphi^*$ due to the dependency structure between the noise and the characteristics of the individuals. In this case removing the bias is equivalent to the estimation of the regression removing the endogeneity. Estimation with endogeneity has been tackled in the econometric literature but also in the biomedical statistics literature, where this phenomenon is known as estimation with confounding variable. In this case, the function $\varphi^*(\cdot)$ is not well-defined and more assumptions are needed to remove the endogenous component. A solution to this problem is commonly obtained by assuming the existence of another source of variability. This is the so-called Instrumental Variables (IV)'s Florens (2003), $\mathbf{W} = (W_1, W_2, \cdots, W_k)$, which need to satisfy two hypotheses described for instance in Carrasco et al. (2014)

1. an independence condition with the noise: $\mathbb{E}[U|\mathbf{W}] = 0$;

2. a sufficiency relation with the assigned treatment

$$\mathbb{E}[\varphi^*(\mathbf{x})|\mathbf{W}] \underset{\text{a.s.}}{=} 0 \Rightarrow \varphi^*(\mathbf{x}) \underset{\text{a.s.}}{=} 0$$

The first condition implies a linear equation characterizing $\varphi^*$:

$$\mathbb{E}[Y|\mathbf{W}] = \mathbb{E}[\varphi^*(\mathbf{x})|\mathbf{W}];$$

and the second condition implies the uniqueness of the solution of this equation.

Actually, we can write the model as

$$\mathbb{E}[Y(\mathbf{x}) - \varphi^*(\mathbf{x})|W_1, W_2, \cdots, W_k] = 0. \tag{3}$$

Defining the operator $T(\cdot) = \mathbb{E}[\cdot|W_1, W_2, \cdots, W_k]$, and the function $r(w)) = \mathbb{E}[Y(\mathbf{x})|\mathbf{W} = w]$, the following equation holds

$$T(\varphi^*) = r \tag{4}$$

Equation (4) models the case where we observe $r$ and we want to estimate $\varphi^*$, which is not observed directly but through an image by an operator $T$. This setting is commonly referred to as an Inverse Problem, see for instance in Engl et al. (1996). There are many ways to solve Inverse Problems in the context of econometrics, as in Carrasco et al. (2007), Loubes & Ludena (2008); Loubes & Marteau (2012), but here we are inspired by recent techniques in this field, and we will learn how to automatically remove the endogeneity effect. The endogeneity can be seen as a bias on the observed information, $Y(\mathbf{x})$, and thus we can promote unbiasedness with this framework. Note that fairness for IV regression has been presented in Centorrino et al. (2022).

Yet in many cases, the required additional information provided by the observation of such instrumental variables is not available. To deal with such cases, the new literature on statistical learning in Inverse Problems, such as Arridge et al. (2019) for instance, provides interesting directions to solve inverse problem and thus post-process bias in supervised Machine Learning. When dealing with Inverse Problems in the usual setting of Machine Learning, it is common to have access to a training set $\{\varphi^*(\mathbf{x}_i), Y(\mathbf{x})_i\}_{i=1}^T$, *i.e.*, a set that associates the samples of the estimated signal to the samples of the reference one, considering a total of $T$ available points. Then deep neural networks are used to invert the observations and estimate an invert operator directly from the data. Such new methods are often referred to as unrolling the inverse problem, see for instance in Monga et al. (2021) and references therein. Yet in our framework, having access to the true unbiased function $\varphi^*$ is an unrealistic setting. Rather, we will use a paradigm of learning called Weakly supervised learning Zhou (2018). In this case, we do not have access to $\{\varphi^*(\mathbf{x}_i), Y(\mathbf{x}_i)\}_{i=1}^T$ for all $T$ available samples, but only for a small fraction of them, namely $\{\varphi^*(\mathbf{x}_i), Y(\mathbf{x}_i)\}$, for $\mathbf{x}_i \in \mathcal{X}_L$, where $\mathcal{X}_L$ represents the smaller set of labeled data. Let denote by $\mathcal{X}_U$ the observations which are unlabeled, hence we have

$$\{1, \ldots, T\} = \mathcal{X}_L \cup \mathcal{X}_U.$$

Usually, the amount of labeled data in this paradigm of learning is very small, $|\mathcal{X}_L| \ll T$, which poses a challenge to the estimation of $\varphi^*(\mathbf{x})$. To circumvent this lack of information, we observe a set of samples of $\varphi^*(\mathbf{x})$, but without establishing the correspondence between these samples and those of the estimated signal. From such a dataset, we will first estimate the probability distribution $\mathbb{P}(\varphi^*)$. The estimation of the distribution of a **fair score** is thus used as a unbiasedness constraint to to better estimate the true fair function $\varphi^*$. Notice that here we replace usual constraint of fairness such as disparate impact, correlation or dependency measures from information theory (described for instance in Oneto & Chiappa (2020) or Chouldechova & Roth (2020), Bird et al. (2020), Del Barrio et al. (2020)) by a distributional constraint. Usually when the sensitive variable is discrete, distributional fairness constraints are built by considering that a fair model should behave in the same way for all subgroups. This is modeled by measuring the distance between the conditional distributions of the model for all subgroups and imposing that all conditional distributions are close to a central distribution. This is the main topic of Jiang et al. (2020), Del Barrio et al. (2019), Risser et al. (2022b) or Gordaliza et al. (2019). Here, the novelty lies also in the fact that we do not specify in advance what does it mean to be fair but rather we learn this fairness constraint from the unbiased sample and reconstruct a fair solution directly under this constraint.

By estimating $\mathbb{P}(\varphi^*)$ and by observing a few training pairs, $\{\varphi^*(\mathbf{x}_i), Y(\mathbf{x}_i)\}$, we propose a procedure able to mitigate the bias of a supervised ML model, inspired by the recent work of Mukherjee et al. (2021), which will be detailed next.

## 3 Methodology

In this section, we first detail our approach based on Inverse Problems to mitigate bias and then, we present a theoretical guarantee that assess its performance.

### 3.1 Bias mitigation in supervised Machine Learning Models

Recall the observation model (1)

$$Y(\mathbf{x}) = \varphi^*(\mathbf{x}) + U(\mathbf{x}).$$

The input characteristics are a continuous variable $\mathbf{x}$ that follows a distribution, $\mathbf{x} \sim \mathbb{P}(\mathbf{x})$. We have modeled $\mathbf{x}$ as a continuous variable to represent attributes such as age, financial status or ethnic proportions, which are known to be potential sources of bias, as in Mary et al. (2019).

In terms of probability distribution, the observed score $Y(\mathbf{x})$ has a probability distribution given by

$$\mathbb{P}(Y) = \mathbb{P}(\varphi^*) * \mathbb{P}(U)$$

where the operator $*$ denotes the convolution operation.

The problem investigated here is the one where a model takes samples of the attributes encoded by $\mathbf{x}$, and generates the scores $Y(\mathbf{x})$. The produced score will, then, be used to select candidates to universities or to job positions, for example.

However, the distribution of such scores may be biased, favoring some individuals to the detriment of others, *e.g.*, younger to the detriment of older people or richer to the detriment of poorer ones. In order to have an unbiased model, or at least one whose bias was mitigated, it is necessary to perform some *a posteriori* treatment over the probability of such scores.

In a weakly supervised learning framework, we assume that we can assign **unbiased** or **fair** scores, denoted by $\varphi^*(\mathbf{x})$, to a few individuals of the population. This assignment could be done, for example, by some surveys or external analysis, which would give us the true ability, *i.e.*, an unbiased/fair score, to a well-chosen set of candidates. This additional information, yet limited to a little number of observations, will be key to mitigate the bias for all individuals.

To better understand our approach to mitigate bias, let us consider the data processing pipeline depicted in Figure 1.

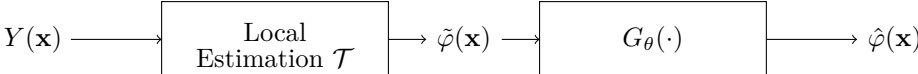

Figure 1: Data processing pipeline: we first perform an initial and simple estimation on $Y(\mathbf{x})$, generating the initial estimate $\tilde{\varphi}(\mathbf{x})$; then, we use a neural network to refine it, producing the final estimate $\hat{\varphi}(\mathbf{x})$.

Our treatment consists of two steps: 1) we observe $Y(\mathbf{x})$ and perform a local estimation $\mathcal{T}(\cdot)$ on it, producing $\tilde{\varphi}(\mathbf{x})$; 2) we take $\tilde{\varphi}(\mathbf{x})$ and further refine it, by using a neural network $G_\theta(\cdot)$, that will produce $\hat{\varphi}(\mathbf{x})$. The role of the first step is to perform an initial estimation, to render the subsequent treatment more robust and stable Genzel et al. (2022). Such a treatment is a naive one, and the initial estimate, $\tilde{\varphi}(\mathbf{x}) = \mathcal{T}(Y)(\mathbf{x})$, may not verify the desired properties. Therefore, we will proceed with step 2, increasing the the performance of the bias mitigation by training a Deep Neural Network (DNN), $G_\theta(\cdot)$, parameterized by $\theta$ and whose

architecture will be described later. The output of such a neural network, $\hat{\varphi}(\mathbf{x})$, is the final estimate, and is desired to be as close as possible to $\varphi^*(\mathbf{x})$.

In an end-to-end point of view, we are performing the following compound operation

$$\hat{\varphi}(\mathbf{x}) = (G_{\hat{\theta}} \circ \mathcal{T})(Y(\mathbf{x})),$$

such that, ideally, we would have a final estimated score $\hat{\varphi}(\mathbf{x})$

- as close as possible to $\varphi^*(\mathbf{x})$ for the few individuals in the population for whom we know the fair scores,

$$(G_\theta \circ \mathcal{T})(Y(\mathbf{x})) = \varphi^*(\mathbf{x}),$$

  for $\mathbf{x} \in \mathcal{X}_L$, a set that will properly defined next;

- whose probability distribution $\mathbb{P}(\hat{\varphi})$ is close to the distribution of the unbiased score $\mathbb{P}(\varphi^*)$. Note that we use the following notation $T_\sharp P$ denoting the push forward operation (see Villani et al. (2009) for instance) which is defined as the image of the measure by the map $T$ as $T_\sharp P = P \circ T^{-1}$.

To accomplish such a task, we will choose a quadratic norm to assess the correspondence between $\hat{\varphi}(\mathbf{x})$ and $\varphi^*(\mathbf{x})$ for $\mathbf{x} \in \mathcal{X}_L$, and a well chosen distance to measure the match between $\mathbb{P}(\varphi^*)$ and $\mathbb{P}(\hat{\varphi})$, to be specified later.

Hence, we propose to train a neural network by choosing the set of parameters that minimizes a cost function composed of two terms, each one minimized over two sets of the data, either the reduced supervised set $\mathcal{X}_L$ or the whole observations $\{1, \ldots, T\}$. The terms can be written as follows:

1. $\mathcal{L}_{\mathcal{X}_L}\big(\varphi^*(\mathbf{x}), G_\theta(\tilde{\varphi}(\mathbf{x}))\big)$, called data-fidelity term, which corresponds to the match of the neural network's output to the unbiased scores. This loss requires the knowledge of the fair scores $\varphi^*$, which are unknown in general, but available at well-chosen points. Hence, we only learn this part on a limited number of observations, belonging to the set $\mathcal{X}_L$. Algebraically we have

$$\mathcal{L}_{\mathcal{X}_L}(\theta) = \sum_{\mathbf{x}_i \in \mathcal{X}_L} \big(\varphi^*(\mathbf{x}_i) - G_\theta(\tilde{\varphi}(\mathbf{x}_i))\big)^2$$

2. $R\big(G_\theta(\tilde{\varphi}(\mathbf{x}))\big)$, called regularization term, which enforces the distribution of the output to be close to the unbiased distribution. The distance chosen to measure the difference between the distributions will be the 1-Wasserstein distance, defined as follows. For two distributions $\mu$ and $\nu$ and for $\Gamma(\mu, \nu)$ the set of joint distributions with marginals $\mu$ and $\nu$, let 1-Wasserstein distance between $\mu$ and $\nu$ be defined as

$$W_1(\mu, \nu) = \inf_{\gamma \in \Gamma(\mu, \nu)} \int \|x - y\| d\gamma(x, y).$$

Note that the distribution of $\varphi^*$ can be computed without knowing which score is biased and which is unbiased, but only considering the data as a whole.

To do so, we numerically estimate the probability distribution $\mathbb{P}(\varphi^*)$ from the data points $\varphi^*(\mathbf{x}_i)$, $i = 1, \cdots, T$, as follows

$$\mathbb{P}(\varphi^*) = \frac{1}{T} \sum_{i=1}^{T} \delta_{\varphi^*(\mathbf{x}_i)}$$

where $\delta(x) = 1$ if $x = 0$ and $\delta(x) = 0$ otherwise.

It is important to note that we do not have access to the training pairs $\{\varphi^*(\mathbf{x}_i), Y(\mathbf{x})_i\}$ for $i = 1, \ldots, T$, as is done in Supervised Learning. Even though we have chosen such an approach

to estimate the reference distribution, any other one could have been used, provided we have access to a suitable estimate of $\mathbb{P}(\varphi^*)$.

We proceed in the same manner to estimate $\mathbb{P}(G_\theta(\tilde{\varphi}))$,

$$\mathbb{P}(G_\theta(\tilde{\varphi})) = \frac{1}{T} \sum_{i=1}^{T} \delta_{\hat{\varphi}(\mathbf{x}_i)}$$

where $\hat{\varphi}(\mathbf{x}_i) = G_\theta(\tilde{\varphi}(\mathbf{x}_i)))$. After estimating the probability distributions, we calculate the regularization term for all $\mathbf{x}_i$ $i = 1, \ldots, T$,

$$R\big(G_\theta(\tilde{\varphi}(\mathbf{x}))\big) = W_1\big(\mathbb{P}(\varphi^*), \mathbb{P}(G_\theta(\tilde{\varphi}))\big).$$

The term $W_1$ denotes the 1-Wasserstein distance between the empirical distributions of the $T$ samples and by using the Sinkhorn algorithm, with $\epsilon = 1.10^{-4}$ Cuturi (2013).

Finally, we consider the regularized loss $L_\lambda(\theta)$, which can be written as the sum between the two previous terms

$$L_\lambda : \theta \longrightarrow \mathcal{L}_{\mathcal{X}_L}(\theta) + \lambda R\big(G_\theta(\tilde{\varphi}(\mathbf{x}))\big). \tag{5}$$

The hyperparameter $\lambda$ controls the trade-off between these two terms, where larger $\lambda$ enforces fairness of the solution by getting its distribution as close as possible, in the sense of the 1-Wasserstein distance, to the so-called *fair distribution*. In all of our simulations, the minimization of the cost function is carried out by algorithms based on the gradient of both terms, and such gradients are automatically calculated using PyTorch Paszke et al. (2017) and GeomLoss Feydy et al. (2019) frameworks.

The regularization term is given by the Wasserstein Distance, which has been used in many works of supervised Machine Learning Frogner et al. (2015); Arjovsky et al. (2017); Mukherjee et al. (2021); Heaton et al. (2022). This term is the responsible for performing the match between the probability distribution of $\varphi^*$ and the one of $G_\theta(\tilde{\varphi})$.

However, only minimizing $W_1\big(\mathbb{P}(\varphi^*), \mathbb{P}(G_\theta(\tilde{\varphi}))\big)$, is not enough to ensure that $\hat{\varphi}(\mathbf{x})$ is close enough to $\varphi^*(\mathbf{x})$. Actually, we could obtain two estimates, $\hat{\varphi}_1(\mathbf{x})$ and $\hat{\varphi}_2(\mathbf{x})$, with their probability distributions as close as possible to the one of $\varphi^*(\mathbf{x})$, but these two estimates may correspond to each other up up to a permutation in their samples. This situation is illustrated in Figure 2.

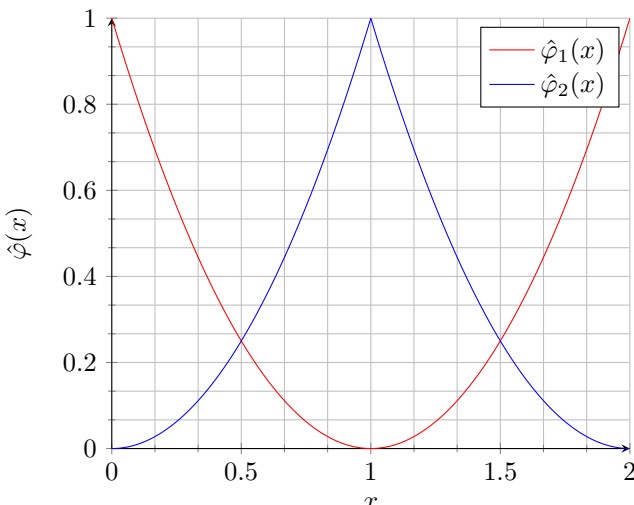

Figure 2: Illustration of the permutation problem that arises from the minimization of the Wasserstein Distance. In both cases, we have the same score values for the red and the blue lines, but they are not properly sorted.

In Figure 2, we illustrate two squared functions (only for simplicity), composed of the same samples (and, therefore, they have the same probability distributions), up to a permutation. Such a permutation may pose a major problem in social applications, since it changes the scores assigned to each individual, and, as a consequence, the decisions taken based on such scores. For example, the maximum score was assigned to the individuals represented by $x = 0$ and $x = 2$ by $\hat{\varphi}_2(\mathbf{x})$, and to the individual represented by $x = 1$ by $\hat{\varphi}_1(\mathbf{x})$, which, in turn, will change who is selected for a job position, for example.

This permutation problem highlights two interesting facts: first, by minimizing the Wasserstein distance, we have found the correct values for $\varphi^*(\mathbf{x})$, but we need to properly sort them; second, such sorting step cannot be accomplished by a procedure based on unsupervised learning, since we need to know which individual should receive a particular score. To solve both of these problems, we employed the so called Weakly Supervised Learning Zhou (2018), illustrated in Figure 3, to complete the training of the neural network. In such approach, from all the available data, we have only a small fraction that was labeled, represented by the set $\mathcal{X}_L$ (blue square), and, hence, can be used in a supervised manner. The other part of the data $\mathcal{X}_U$, represented by the gray area in Figure 3, was not labeled and should be used in an unsupervised manner, which we have done with the Wasserstein distance. In the simulations, we will vary this proportion of known data.

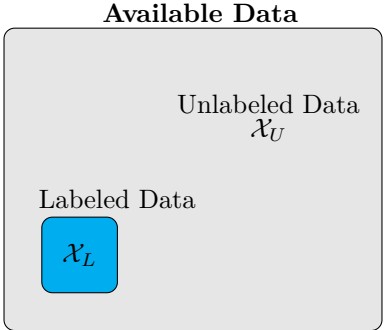

Figure 3: Weakly supervised learning: from all of the available data, only a small fraction has been labeled, as represented by the blue area. The remaining data — gray area — must be used in an unsupervised manner.

Hence, our approach here is to use a small fraction of labeled data to complete the estimation process. This is the role of the term $\mathcal{L}_{\text{labeled}}\big(\varphi^*(\mathbf{x}), G_\theta(\tilde{\varphi}(\mathbf{x}))\big)$ in equation (5): we can link samples from $Y(\mathbf{x})$ to the samples of $\varphi^*(\mathbf{x})$, by using a few training pairs in $\mathcal{X}_L$ (typically $|\mathcal{X}_L| \ll T$), to avoid the permutation issue. In the context of our work, where we observe scores obtained by individuals through a possibly biased treatment, this is equivalent to performing a polling on some individuals of the population, analysing their characteristics, and, then, attributing to them the scores that they would deserve, which is assimilated to the unbiased scores. This framework is similar to the ideas developed in Friedler et al. (2021) and correspond to values in a construct space where unbiased versions are available, opposed to the observations or the decisions which reflect the biases of our world or the biases of the algorithmic decisions. Having access to the fair scores requires an analysis that, for societal applications, cannot be done for all individuals but is limited to a few cases.

In simple words, the approach that we use here to mitigate bias consists of two steps: first, we need to know the correct values that $\hat{\varphi}(\mathbf{x})$ should have for all the individuals (which we achieve in an unsupervised manner, by minimizing the Wasserstein distance); second, we need to properly sort these values (which we achieve by obtaining the correspondence between $\hat{\varphi}(\mathbf{x})$ and $\varphi^*(\mathbf{x})$ for the few known training pairs).

Now that we have described our bias mitigation approach, we will present the theoretical analysis that provides some bounds of its performance.

### 3.2 Theoretical Guarantees

The following theorem states that a minimizer of (5) has its performance bounded by the performance of "specialists", *i.e.* models that were trained only to minimize either $\mathcal{L}_{\text{labeled}}\big(\varphi^*(\mathbf{x}), G_\theta(\tilde{\varphi}(\mathbf{x}))\big)$ or $R\big(G_\theta(\tilde{\varphi}(\mathbf{x}))\big)$, separately.

**Theorem 1.** *Let us consider the following cost function to be minimized*

$$J(G_\theta|\lambda) = \sum_{i \in \mathcal{X}_L} \big(\varphi^*(\mathbf{x}_i) - G_\theta(\tilde{\varphi}(\mathbf{x}_i))\big)^2 + \lambda W_1\big(\mathbb{P}(\varphi^*), \mathbb{P}(G_\theta(\tilde{\varphi}))\big),$$

*and the sets*

$$\Theta_L = \left\{ \theta : \sum_{i \in \mathcal{X}_L} \big(\varphi^*(\mathbf{x}_i) - G_\theta(\tilde{\varphi}(\mathbf{x}_i))\big)^2 = 0 \right\},$$

$$\Theta_W = \left\{ \theta : (G_\theta)_\sharp \mathbb{P}(\tilde{\varphi}) = \mathbb{P}(\varphi^*) \right\}.$$

*Let $G_{\theta^*}$ be a minimizer of $J(\cdot|\lambda)$. Then it holds that for all $\theta \in \Theta_L$,*

$$\sum_{i \in \mathcal{X}_L} \big(\varphi^*(\mathbf{x}_i) - G_{\theta^*}(\tilde{\varphi}(\mathbf{x}_i))\big)^2 \leq \lambda W_1\big(\mathbb{P}(\varphi^*), \mathbb{P}(G_\theta(\tilde{\varphi}))\big),$$

*and for all $\theta \in \Theta_W$,*

$$W_1\big(\mathbb{P}(\varphi^*), \mathbb{P}(G_{\theta^*}(\tilde{\varphi}))\big) \leq \frac{1}{\lambda} \sum_{i \in \mathcal{X}_L} \big(\varphi^*(\mathbf{x}_i) - G_\theta(\tilde{\varphi}(\mathbf{x}_i))\big)^2.$$

Hence, the theorem states that we can establish bounds for a minimizer of $J_2$ by using the performance of "specialists", *i.e.*, models that were trained to only minimize one of the terms of $J_2$. A minimizer $G_{\theta^*}$ will have a better performance in terms of data fidelity than a data-fidelity specialist used to minimize the Wasserstein distance; in a dual manner, a minimizer $G_{\theta^*}$ will have its regularization perform upper-bounded by the performance of a regularization-specialist applied to the data-quality term.

This result is inspired by the work of Mukherjee et al. (2021) where the authors propose an adversarial approach to solve Inverse Problems, in the context of image analysis for a model

$$\mathbf{y}^\delta = A(\mathbf{x}) + \epsilon \tag{6}$$

where $A(\cdot)$ is the forward operator, $\mathbf{y}^\delta$ are the noisy measurements and $\epsilon$, $||\epsilon||_2 \leq \delta$, is the noise. Hence, *mutatis mutandis* the analysis made in Mukherjee et al. (2021) in **Proposition 1**, we obtain the proof of the previous theorem.

*Proof.* If $G_{\theta^*}$ is a minimizer of $J(\cdot|\lambda)$, then for every $\theta \in \Theta_L$

$$\sum_{i \in \mathcal{X}_L} \big(\varphi^*(\mathbf{x}_i) - G_{\theta^*}(\tilde{\varphi}(\mathbf{x}_i))\big)^2 + \lambda W_1\big(\mathbb{P}(\varphi^*), \mathbb{P}(G_{\theta^*}(\tilde{\varphi}))\big) \leq \sum_{i \in \mathcal{X}_L} \big(\varphi^*(\mathbf{x}_i) - G_\theta(\tilde{\varphi}(\mathbf{x}_i))\big)^2 +$$
$$+ \lambda W_1\big(\mathbb{P}(\varphi^*), \mathbb{P}(G_\theta(\tilde{\varphi}))\big).$$

Naturally, we have

$$\sum_{i \in \mathcal{X}_L} \big(\varphi^*(\mathbf{x}_i) - G_{\theta^*}(\tilde{\varphi}(\mathbf{x}_i))\big)^2 \leq \sum_{i \in \mathcal{X}_L} \big(\varphi^*(\mathbf{x}_i) - G_{\theta}(\tilde{\varphi}(\mathbf{x}_i))\big)^2 +$$
$$+ \lambda W_1\big(\mathbb{P}(\varphi^*), \mathbb{P}(G_{\theta}(\tilde{\varphi}))\big) - \lambda W_1\big(\mathbb{P}(\varphi^*), \mathbb{P}(G_{\theta^*}(\tilde{\varphi}))\big).$$

Since $\theta \in \Theta_L$, $\sum_{i \in \mathcal{X}_L} \big(\varphi^*(\mathbf{x}_i) - G_{\theta}(\tilde{\varphi}(\mathbf{x}_i))\big)^2 = 0$, leading to

$$\sum_{i \in \mathcal{X}_L} \big(\varphi^*(\mathbf{x}_i) - G_{\theta^*}(\tilde{\varphi}(\mathbf{x}_i))\big)^2 \leq \lambda W_1\big(\mathbb{P}(\varphi^*), \mathbb{P}(G_{\theta}(\tilde{\varphi}))\big) - \lambda W_1\big(\mathbb{P}(\varphi^*), \mathbb{P}(G_{\theta^*}(\tilde{\varphi}))\big).$$

By using the fact that $W_1(\cdot, \cdot) \geq 0$, we finally have

$$\sum_{i \in \mathcal{X}_L} \big(\varphi^*(\mathbf{x}_i) - G_{\theta^*}(\tilde{\varphi}(\mathbf{x}_i))\big)^2 \leq \lambda W_1\big(\mathbb{P}(\varphi^*), \mathbb{P}(G_{\theta}(\tilde{\varphi}))\big), \ \forall \theta \in \Theta_L.$$

Analogously, for every $\theta \in \Theta_W$, we have

$$\lambda W_1\big(\mathbb{P}(\varphi^*), \mathbb{P}(G_{\theta^*}(\tilde{\varphi}))\big) \leq \sum_{i \in \mathcal{X}_L} \big(\varphi^*(\mathbf{x}_i) - G_{\theta}(\tilde{\varphi}(\mathbf{x}_i))\big)^2 - \sum_{i \in \mathcal{X}_L} \big(\varphi^*(\mathbf{x}_i) - G_{\theta^*}(\tilde{\varphi}(\mathbf{x}_i))\big)^2 +$$
$$+ \lambda W_1\big(\mathbb{P}(\varphi^*), \mathbb{P}(G_{\theta}(\tilde{\varphi}))\big)$$

Using the fact that $\theta \in \Theta_W$ and $W_1\big(\mathbb{P}(\varphi^*), \mathbb{P}(G_{\theta}(\tilde{\varphi}))\big) = 0$, and the non-negativity of the other terms, we have

$$\lambda W_1\big(\mathbb{P}(\varphi^*), \mathbb{P}(G_{\theta^*}(\tilde{\varphi}))\big) \leq \sum_{i \in \mathcal{X}_L} \big(\varphi^*(\mathbf{x}_i) - G_{\theta}(\tilde{\varphi}(\mathbf{x}_i))\big)^2, \forall \theta \in \Theta_W.$$

$\square$

Having described our approach and theoretically analyzed it, in the next section we will evaluate it by means of numerical simulations.

## 4    Numerical Simulations

To evaluate our approach, we will consider in the following numerical simulations 1- and 2-dimensional signals. Revisiting the example presented in the introduction, about risk assignment made by assurance companies, in the first case we observe the risk score $Y(x)$, produced by only taking into account a single variable $x$ (that could represent financial status, for example), and we know that it is biased, *i.e.*, for specific values of $x$, we would observe more frequently the occurrence of high risk scores. In this case, the bias term $U(x)$ is modeled as noise whose mean is proportional to $x$, representing a scenario of a more controlled bias. For the 2-dimensional case, we observe the risk score $Y(x_1, x_2)$ that now depends on two attributes (financial status and age, for example) and is also biased. For the 2-D case, we propose a more challenging bias term, allowing the noise to depend on $x_1$, $x_2$ and also on the product $x_1 x_2$. The functions we used to model the unbiased score term, $\varphi^*(x)$ in the 1-dimensional case and $\varphi^*(x_1, x_2)$ in the 2-dimensional case, are inspired by real use cases in econometrics. For each simulation set, we provide and discuss the architecture of the neural network and the choice for hyperparameters.

We also perform some numerical simulations to assess the relation between the model's performance (measured by the Mean Squared Error between the estimated unbiased score and the true one) and the number of training data points labeled. Finally, since the concept of fairness is a very complex one, and what is considered to be fair today may not be considered to be fair in the future, we also performed some numerical simulations to evaluate capacity of the proposed method to track the unbiased function over time.

### 4.1  1-Dimensional Signals

For the 1-dimensional case, we generated $T = 1000$ uniformly spaced samples for $x$ in the interval [-3, 3], and $\varphi(x) = x^2$ Mas-Colell et al. (1995). To generate the bias term in equation (2), we generate a noise with mean

$$\mu(x) = \alpha x,$$

with $\alpha = 2$, and variance $\sigma(x) = 1$. Since the noise is correlated with $x$, it would affect systematically differently the various subgroups of the population (*i.e.*, the different partitions that one may have on $x$), leading to a biased score.

For comparative purposes, we have employed the so-called Instrumental Variables (IVs) to debias $Y(x)$. To suitably generate the IVs, we used the following procedure Florens (2003):

1. For a number $k$ of IVs, we generate the temporary variable $e = (e_1, \cdots, e_k)'$, from a standard uniform distribution;

2. For each $j = 1, \cdots, k$, we have

$$\epsilon_j = \sqrt{\frac{k}{2}} \frac{e_j}{\sum_{j=1}^{k} e_j}, \quad \tau_j = \frac{1}{j} \sum_{l=1}^{j} e_j$$

3. The instrumental variables are generated as

$$W \equiv w(\tau_j) = \Phi^{-1}\big(\Phi(1) + \tau_j(\Phi(1) - \Phi(-1))\big)\sigma$$

$\sigma = 1.853$, so that $w(\tau_j)$ follows a truncated normal distribution between $[-\sigma, \sigma]$, for each $j = 1, \cdots, k$.

In (4), the operator $T$ is approximated by a local linear non-parametric regression, whose bandwidth is adjusted by promoting stability of the changes in the accuracy. We did the same for the adjoint operator, $T^*(\cdot) = \mathbb{E}[\cdot|X]$.

Having access to suitable approximations to $T$ and $T^*$, we can now solve (4) by using the Landweber-Fridman algorithm Centorrino & Florens (2021):

$$\hat{\varphi}_{i+1} = \hat{\varphi}_i + cT^*(T\hat{\varphi}_i - r), \ i = 1, \cdots, N, \tag{7}$$

where $N$ is the regularization parameter (chosen by leave-one-out cross validation), which controls the number of iteration, and $c \in (0, 1)$ is a constant to avoid instability issues (we used $c = 0.5$).

We present the results for $k = 2, 10, 25$ IVs in Figures 4, 5 and 6, respectively, with the associated $\ell_2$ norm of the estimation error, $||\varphi^* - \hat{\varphi}||_2$. In those figures, the left plot represents the true data, *i.e.*, the fair score; the plot in the middle represents the observed data, *i.e.*, the biased score that must be correct, and in the left plot we have a comparison of the true score (blue line) with the estimated one (orange).

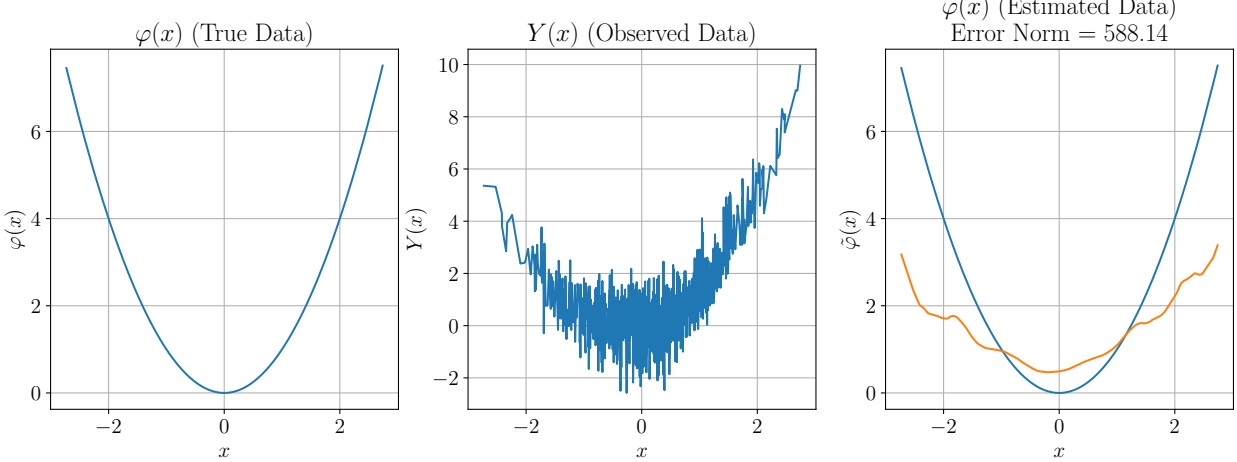

Figure 4: Instrumental Regression and Landweber Iteration - $k = 2$.

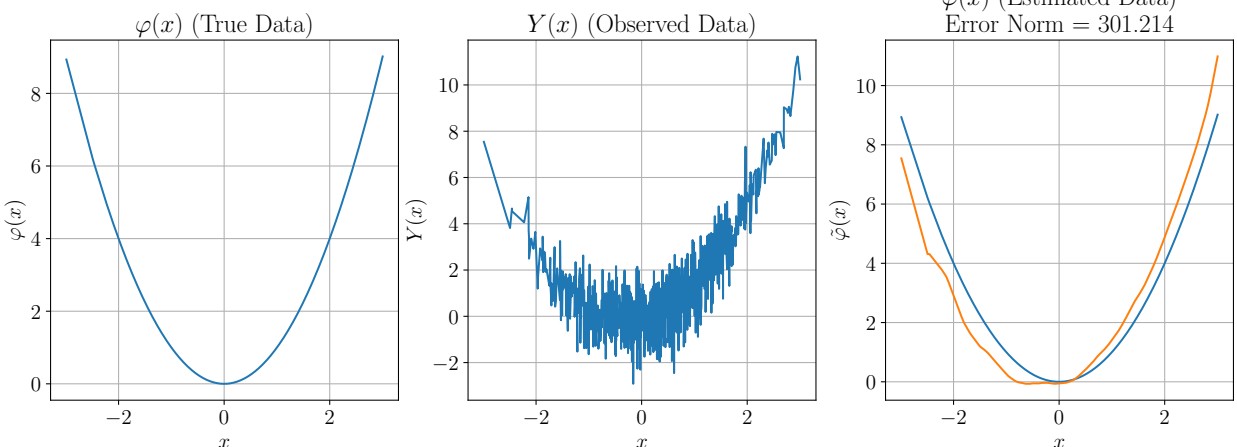

Figure 5: Instrumental Regression and Landweber Iteration - $k = 10$.

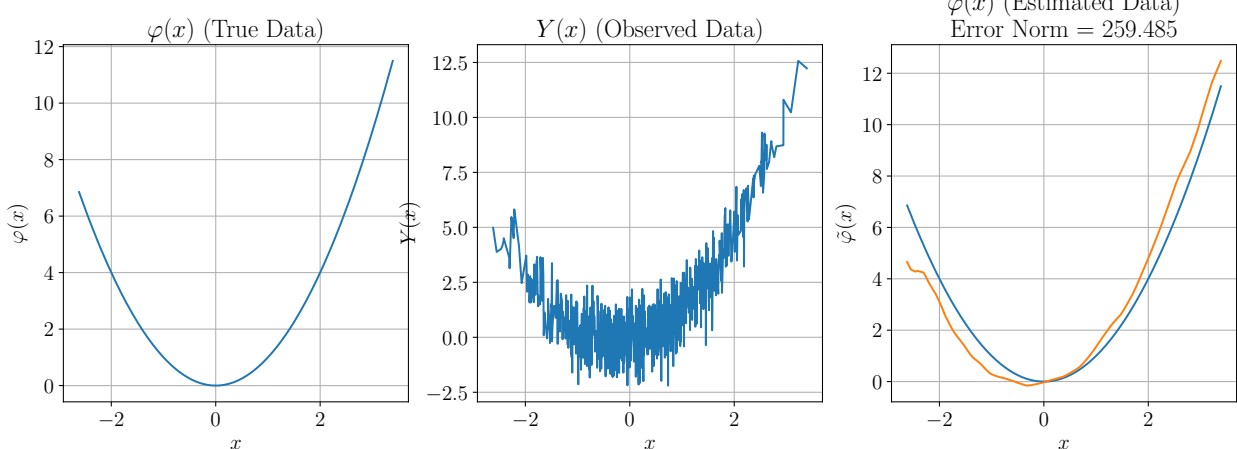

Figure 6: Instrumental Regression and Landweber Iteration - $k = 25$.

From the above results, we note that $k = 2$ IVs led to an estimate very different from the desired signal, indicating that this number of IVs was not enough to deal with the bias term $U(x)$. To circumvent this issue, we added more IVs to our model, and for $k = 10$ and $k = 25$, we have gotten more precise estimates, even though there is still room for improvement.

Despite the fact that the estimation using $k = 10$ and $k = 25$ IVs produced interesting results, this kind of procedure is expensive in real-world applications: besides the data already collected in $\mathbf{x}$, we would have to collect the complementary information needed by the IVs.

As an alternative to regression IV, we performed the estimation process proposed in Figure 1. To do so, we first used a Moving Average filter, with 10 taps, for the local estimation. Since this procedure only gives an initial estimated, $\tilde{\varphi}(x)$ that is not accurate enough (as we will present in the results), we also used a neural network to complete the estimation.

The used neural network is depicted in Figure 7. It is a fully connected neural network, whose linear layers are composed of 1000 neurons each, and the operation ReLU,

$$\mathrm{ReLU}(x) = \max(0, x),$$

is taken element-wise. We feed $\tilde{\varphi}(x)$ into the neural network, and then we apply twice the transformation composed by a linear layer followed by a ReLU operation. Finally, we use one more time a linear layer to obtain the final estimate, $\hat{\varphi}(x)$.

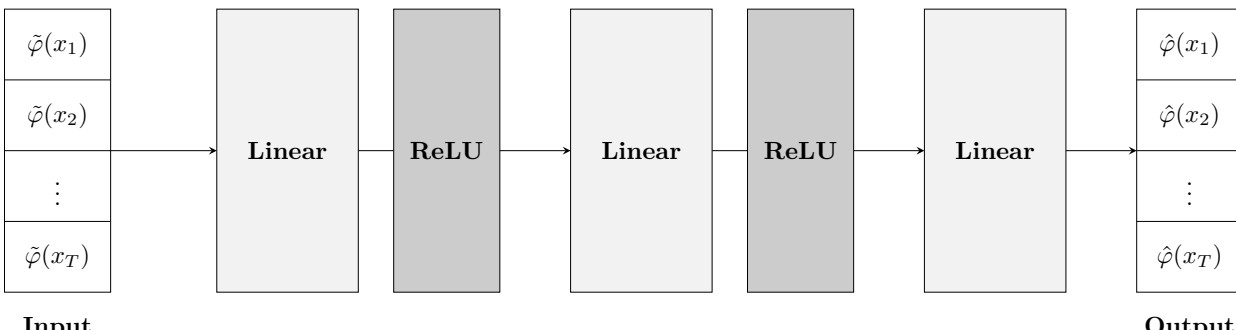

Figure 7: Architecture of the neural network used to treat 1-D signals. It is a fully connected neural network, each linear layer has 1000 neurons and the operator ReLU is taken element-wise.

In this first experiment, we only minimized the Wasserstein distance in (5), *i.e.*, we did not use any labeled data, $|\mathcal{X}_L| = 0$. To properly solve the optimization problem at hand, we used the Adam optimizer Kingma & Ba (2017), with learning rate $\mu = 1.10^{-5}$, for 300 epochs. We present the results for the training dataset in Figure 8. As we did before, we first present the true data, then the observed one (first two plots, from left to right); in the third plot, we present the initial estimate, *i.e.*, the one obtained with the Moving Average filter (orange line represents the true data and blue line represents the estimated one); finally, the fourth plot, depicts the final estimate, the obtained at the output of the used neural network.

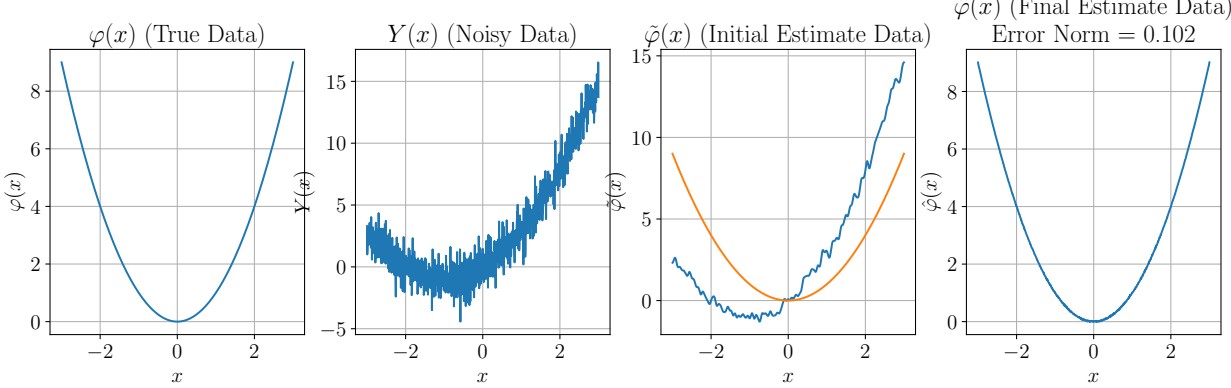

Figure 8: Training dataset, 1-dimensional signals. From left to right: true data $\varphi^*(x)$, observed data $Y(x)$, initial estimate $\tilde{\varphi}(x)$ and final estimate $\hat{\varphi}(x)$.

From Figure 8, we note that the initial estimate, $\tilde{\varphi}(x)$, is less affected by the noise, but it is still distant from the desired signal. After feeding $\tilde{\varphi}(x)$ into the neural network, we got the estimate presented in the fourth figure from left to right. It is a very precise estimate, as can be verified both visually (since we cannot observe a distinction between the orange line for the true data and the blue line for the estimated one) and by the low value of the error norm.

Since we obtained a very good result for the training dataset, we evaluated the trained model in a test dataset. To generate such a dataset, we once again generated $T = 1000$ uniformly spaced samples for $x \in [-3 + \epsilon, 3 + \epsilon]$, where $\epsilon \sim U(-0.5, 0.5)$ and $\varphi(x) = x^2$. By doing so, our test dataset corresponds to a perturbed version of the training one, which is very interesting to assess the generalization of the model. We present the obtained results in Figure 9.

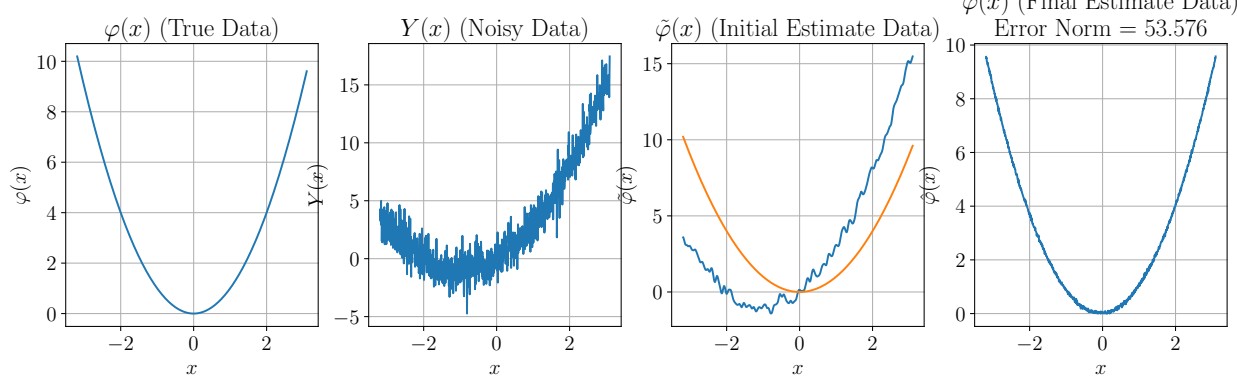

Figure 9: Test dataset, 1-dimensional signals. From left to right: true data $\varphi^*(x)$, observed data $Y(x)$, initial estimate $\tilde{\varphi}(x)$ and final estimate $\hat{\varphi}(x)$.

From Figure 9 we note a performance for the test set very similar to the one obtained with the training one: our first estimate is less affected by the noise, but is still very different from the desired signal. After using the neural network, though, we got a very precise estimate, with an error norm of 53.576 taken in 1000 samples (which gives us an Mean Square Error about 0.005).

The objective with the 1-Dimensional Signals was to evaluate how the proposed method performed when dealing with only one continuous-valued sensitive attribute. As we can infer from the presented results, we obtained a better performance with our approach than that obtained by using Instrumental Variables, even when the number of such variables was considerably high ($k = 25$, for example). It is interesting to note that in this first experiment, we obtained such a good performance by only minimizing the Wasserstein distance, because the initial estimate, $\tilde{\varphi}(x)$, was close enough to the true solution, being necessary only to further regularize it. In more challenging scenarios, this could not be the case, and we would have to use a few labeled data points, as we will illustrate in the next section with 2-dimensional signals.

### 4.2   2-Dimensional Signals

In this section we evaluate our approach with 2-dimensional signals, *i.e.*, the case where $\mathbf{x} = (x_1, x_2)$. We generated $T_1 = 100$ uniformly spaced samples for $x_1$ in the interval [-3, 3] and $T_2 = 100$ uniformly spaced samples for $x_2$ in the interval [-3, 3] (so $\varphi^*(x_1, x_2)$ has $T = 10^4$ samples), and considered the function $\varphi^*(x_1, x_2) = (|x_1|^p + |x_2|^p)^{1/p}$, $p = 2$ Mas-Colell et al. (1995). As for the bias term, we used in (2) a noise with mean

$$\mu(x_1, x_2) = \alpha_1 x_1 + \beta_1 x_2 + \gamma_1 x_1 x_2, \ \alpha = 0.2, \beta = 0.2, \gamma = 1$$

and variance

$$\sigma(x_1, x_2) = \alpha_2 |x_1| + \beta_2 |x_2| + \gamma_2 |x_1|.|x_2|, \ \alpha_2 = 0.5, \beta_2 = 0.5, \gamma_2 = 0.2.$$

In this second scenario, we have a more challenging noise, because it is now correlated with both variables (rendering both $x_1$ and $x_2$ sensitive attributes), but also with their product. This dependence on the product of sensitive variables is relevant for modeling real-life situations where the same individual may experience biases from two different causes, such as age and financial condition.

Since we now have a 2-dimensional signal, we performed the estimation by using techniques that are common to the image processing field. First, we used a Gaussian kernel for the local estimation, with standard deviation equal to 5. Then, we used the neural network depicted in Figure 10.

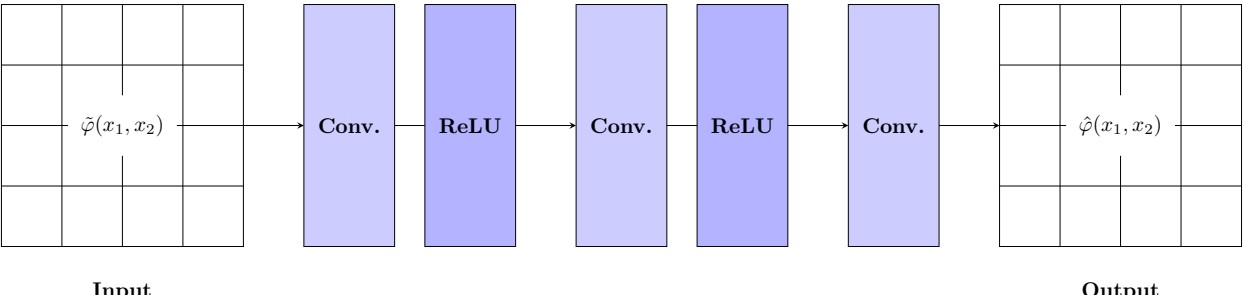

Figure 10: Architecture of the neural network used to treat the 2-D signals. Each convolutional layer is a squared kernel of dimension $100 \times 100$ (the same size as the input and the output) and the operator ReLU is taken element-wise.

We present the initial estimate, $\tilde{\varphi}(x_1, x_2)$, to the neural network, and, then, we process it by using twice in a row a convolutional layer, made of a squared kernel of dimension $100 \times 100$, followed by the ReLU

operation, taken element-wise. To produce the final estimate, $\hat{\varphi}(x_1, x_2)$ we apply a final convolutional layer, with the same dimension as before. Once again, the optimization procedure in (5) was carried out by the Adam optimizer, with $\mu = 1.10^{-5}$ and 12000 epochs.

As we did in the 1-dimensional case, we first applied the local estimator, producing $\tilde{\varphi}(x_1, x_2)$, as depicted in Figure 11, but, once again, we need to further process it in order to get a more precise estimate.

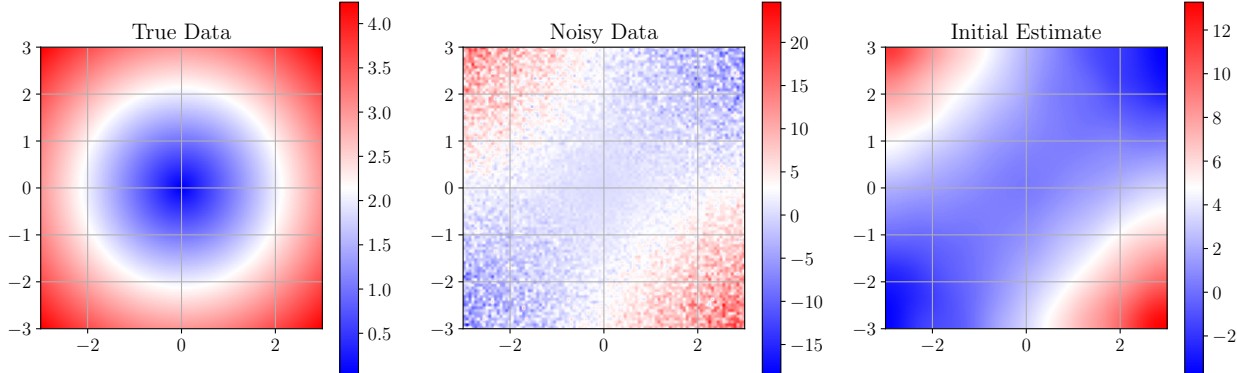

Figure 11: Initial estimation of a 2-dimensional signal. A local estimator is not enough to properly recover the true data.

We first continued the estimation process by considering only the Wasserstein distance, and we present the obtained result in Figure 12.

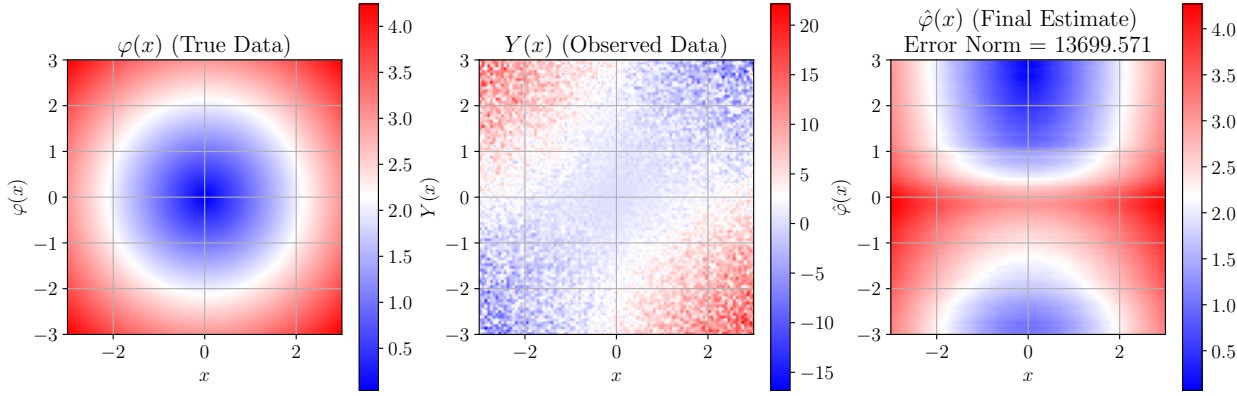

Figure 12: 2-dimensional signals. Estimation without using the labeled data lead to a permutation problem. From left to right: true data $\varphi^*(x_1, x_2)$, observed data $Y(x_1, x_2)$ and final estimate $\hat{\varphi}(x_1, x_2)$.

Here, we have founded the appropriate values for $\hat{\varphi}(x_1, x_2)$, but we could not suitably distribute them (comparing the reference image with the estimated one, the upper and lower parts were permuted). To circumvent this issue, we performed the estimation procedure once again, with the same neural network and the same hyper-parameters, but this time we have used a few training pairs, i.i.d and uniformly selected. In Figure 13 we present the estimated image, after using a number of training pairs that corresponds to 1.0% of all the available data ($|\mathcal{X}_L| = T/100$) and $\lambda = 1.0$ in (5).

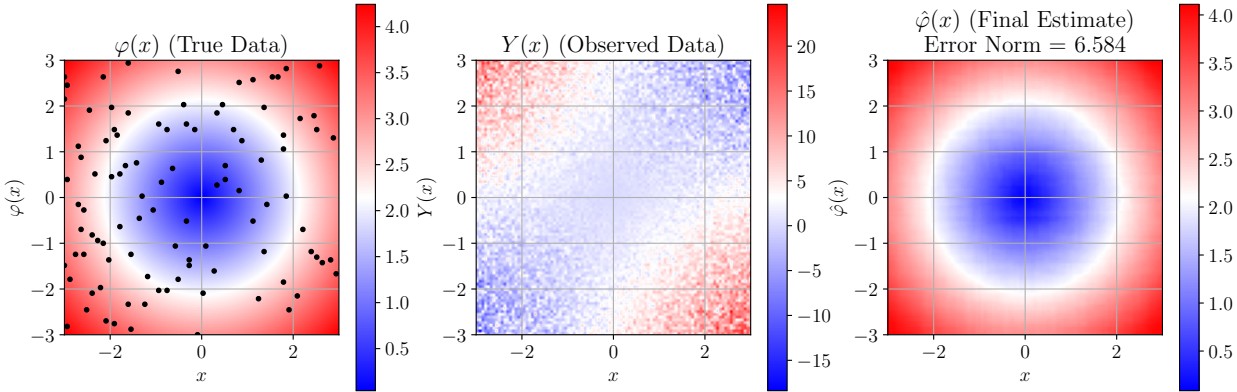

Figure 13: Training dataset, 2-dimensional signals. Number of training pairs: $|\mathcal{X}_L| = T/100$. The black dots in the first image represent the labeled data used in the experiment. From left to right: true data $\varphi^*(x_1, x_2)$, observed data $Y(x_1, x_2)$ and final estimate $\hat{\varphi}(x_1, x_2)$.

By using a very small amount of labeled data, we have obtained a very precise estimate of $\varphi^*(x_1, x_2)$, with error norm of 6.584, and the associated Mean Squared Error (MSE), considering the $10^4$ samples, about $6.5 \times 10^{-4}$.

To better evaluate the performance of such a model, we evaluated its performance on a test dataset. As in the 1-D case, the test set consists in a perturbed version of the training one, where we generated $T_1 = 100$ samples of $x_1$ taken in the interval $[-3 + \epsilon, 3 + \epsilon]$, and $T_2 = 100$ samples of $x_2$ taken in the same interval, with $\epsilon \sim U(-0.5, 0.5)$. We present the obtained results in Figure 14.

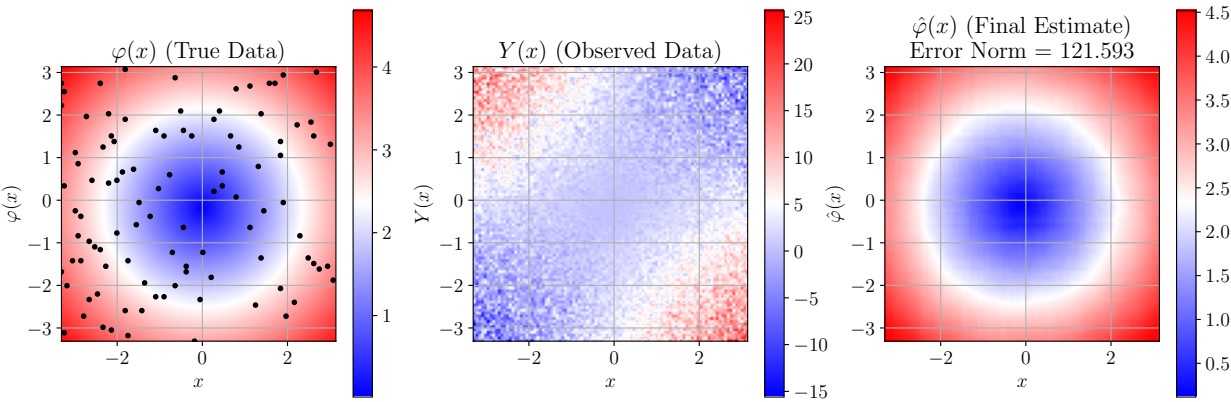

Figure 14: Test dataset, 2-dimensional signals. Number of training pairs: $|\mathcal{X}_L| = T/100$. The black dots in the first image represent the labeled data used in the experiment. From left to right: true data $\varphi^*(x_1, x_2)$, observed data $Y(x_1, x_2)$ and final estimate $\hat{\varphi}(x_1, x_2)$.

In the test dataset, we observe, again, a very precise estimate, with an error norm of 121.593 (and an MSE about 0.01), which indicates that the trained model has a good capacity of generalization. It is important to note that we have used the same amount of training pairs to get such an interesting result.

To further assess the capacity of the proposed debiasing method, we have also considered another function, $\varphi^*(x_1, x_2) = \sin(x_1^2) + \cos(x_2^2)$. This new function is a composition of oscillatory functions, sinus and cosines, and monomials, represented by the squared function. Such a composition poses a more challenging scenario than the previous one.

As we did in the previous case, we performed the local estimation with the same Gaussian kernel, and we got the results presented in Figure 15. Once again, the local estimator reduced the noise effect, but was not able to completely restore the desired signal.

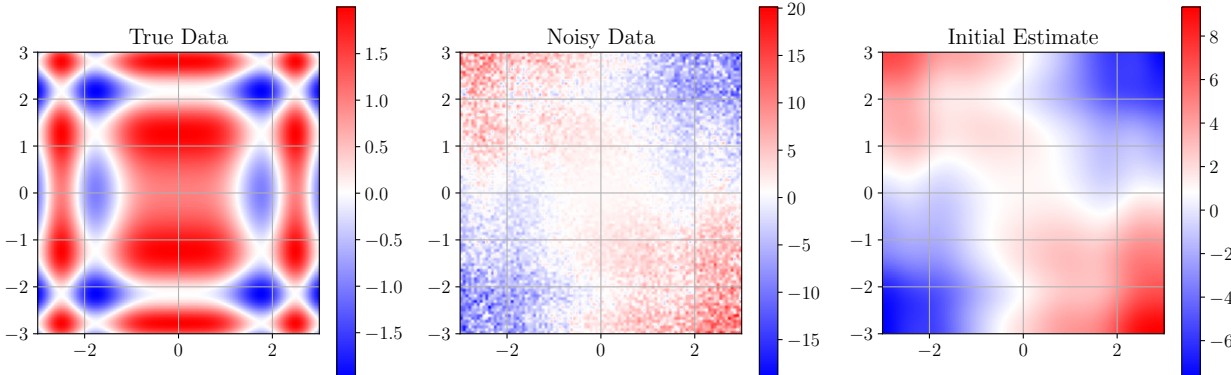

Figure 15: Initial estimation of the second 2-dimensional signal evaluated. Once again, the local estimator is not enough to recover the true data.

We continued the estimation process by only minimizing the Wasserstein distance between the probability distribution of the model's output and the reference one. As can be seen in Figure 16, we could not properly estimate the desired signal with such a procedure, observing, once again, the permutation on the estimated samples.

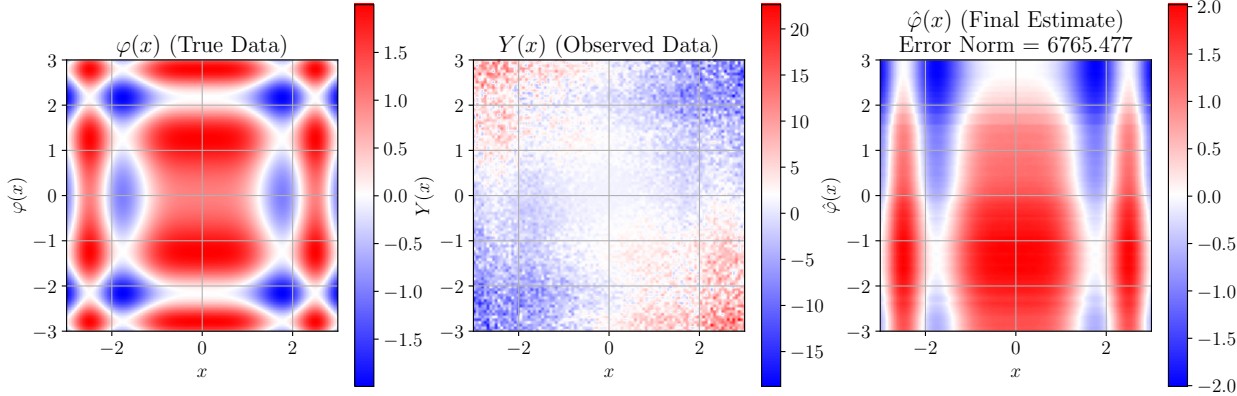

Figure 16: Another example of a 2-dimensional signal, estimated by only using the Wasserstein distance. Again, we observe the permutation problem: we found the true scores values, but we could not assign them correctly.

Hence, to properly estimate the signal, we once again used a few labeled data points, again with $|\mathcal{X}_L| = T/100$ and $\lambda = 1.0$. In Figure 17, we present the obtained result for the training dataset. As we can note, we got an error norm of 60.782, which leads to an MSE about $6.1 \times 10^{-3}$, indicating a very precise estimation.

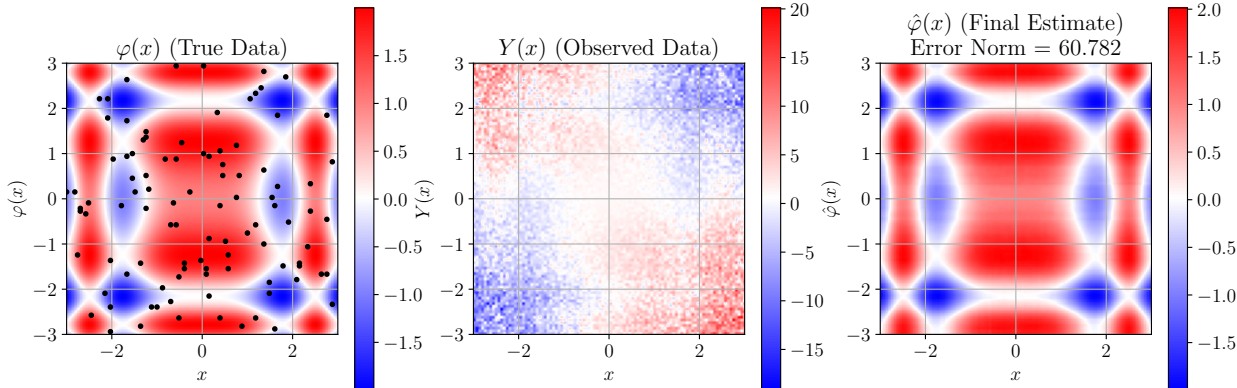

Figure 17: Training dataset, second 2-dimensional signal with $|\mathcal{X}_L| = T/100$. From left to right: true data $\varphi^*(x_1, x_2)$, observed data $Y(x_1, x_2)$ and final estimate $\hat{\varphi}(x_1, x_2)$.

We also evaluated this model using a test dataset, generated in the same manner as in the previous 2-D case; we present the obtained result in Figure 18.

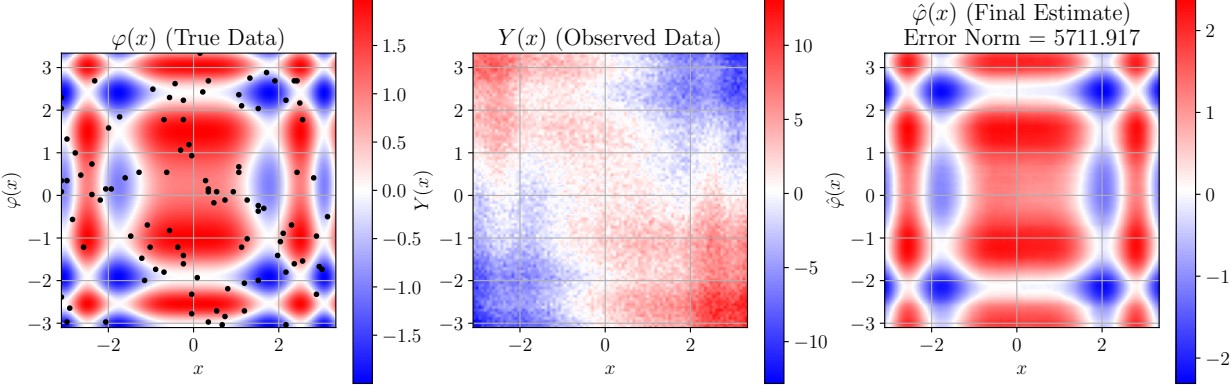

Figure 18: Test dataset, second 2-dimensional signal with $|\mathcal{X}_L| = T/100$. From left to right: true data $\varphi^*(x_1, x_2)$, observed data $Y(x_1, x_2)$ and final estimate $\hat{\varphi}(x_1, x_2)$.

From the results depicted in Figure 18, we observe, once again, a very precise estimate, with error norm of 5711.917 (and an MSE about 0.52), despite the more complex nature of this second function.

Our objective with the 2-Dimensional signals was to evaluate how the proposed method would perform in a more challenging scenario, where we have now two real-valued sensitive attributes, each one being a source of bias, but also their combination can be a source of bias. From the obtained results, we note that only a small amount of labeled data (here, 1.0% of all the samples), alongside a distributional constraint given by the Wasserstein distance, was sufficient to produce very precise estimates, mitigating the bias. It is important to note that we have randomly collected labeled samples from the all available data, indicating that most of the work was done by the regularization term, in an unsupervised manner. Another very interesting point is that the investigation with 2-dimensional signals highlighted the required steps to mitigate the bias in a model: first, it is necessary to find the unbiased score values and, then, to properly distribute them.

**Remark.** *It is important to note that an implicit prior information is encoded into the architecture of the neural network. Here, we have used the ReLU function (Agarap (2018)) as an activation function. This activation function assumes that the signals to be approximated are piece-wise linear, at least in a small neighborhood, or can be well approximated by such linear signals. This fact is very interesting in the context of using few labeled samples, since by knowing the value of the true function in a point, the neural network*

*can estimate, with a good precision, the values of the other points around. This is the reason why, since the economics functions we consider satisfy such assumptions, only a small amount of labeled data is required in this work to yet achieve a good approximation. For less smooth functions, for instance with several discontinuities, it could be necessary to use an amount of data considerably higher than the amount that we used here.*

Finally, to better assess the performance of the proposed approach, we present in Figure 19 the evolution of the Mean Squared Error (MSE) as we increase the percentage of available training pairs, from 0.0% to 1.0%. We also present in Table 1 the mean MSE and the standard deviation, both for the training and the test phase. In all cases, we considered 10 independent realizations of the experiment.

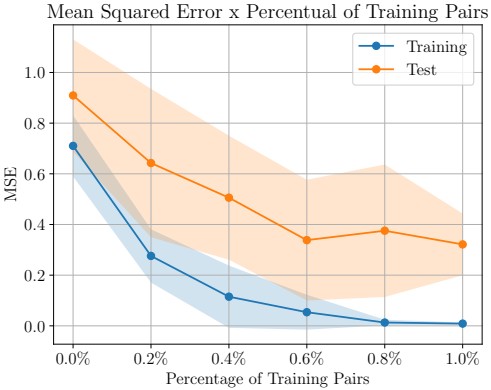

| Perc. Train. Pairs | MSE Train | MSE Test |
|:---:|:---:|:---:|
| 0.0% | $0.710 \pm 0.121$ | $0.909 \pm 0.221$ |
| 0.2% | $0.276 \pm 0.105$ | $0.642 \pm 0.293$ |
| 0.4% | $0.115 \pm 0.123$ | $0.506 \pm 0.245$ |
| 0.6% | $0.054 \pm 0.069$ | $0.338 \pm 0.238$ |
| 0.8% | $0.013 \pm 0.011$ | $0.375 \pm 0.261$ |
| 1.0% | $0.009 \pm 0.005$ | $0.321 \pm 0.122$ |

Table 1: Mean Squared Error vs. the percentage of training pairs. For each line, we present the mean value $\pm$ the standard deviation, obtained from 10 independent realizations of the experiment.

Figure 19: MSE vs. percentage of training pairs, with the mean values (solid lines) and standard deviation (shadowed area) taken from 10 independent realizations.

As we can note both from Figure 19 and Table 1, as we increase the number of available training pairs, the MSE drops down, as expected, and we can note such a behavior for the training and the test phases. Another interesting fact is that as we increase the percentage of training pairs, the variance of the results lowers, as we can observe from the narrower shaded areas, mostly in the training phase. This result show, once again, that a very little amount of supervised data (here, 1.0% or less) can be used to increase the model's performance and to stabilize it, complementing the information encoded in the distributional constraint.

## 4.3 Time-Variant Fairness

Since the concept of fairness is a very complex one and may change over time, its important for fairness-enhancing model to be able to track those time changes, as is the case of our model. To evaluate such a feature, we generated 4 signals that represent the fair score changing over time, as depicted below, with their respective equations.

$$\varphi_1(x_1, x_2) = 0.25 \sin(x_1^2) + 1.75 \cos(x_2^2)$$
$$\varphi_2(x_1, x_2) = 0.5 \sin(x_1^2) + 1.5 \cos(x_2^2)$$
$$\varphi_3(x_1, x_2) = \sin(x_1^2) + 1.5 \cos(x_2^2)$$
$$\varphi_4(x_1, x_2) = \sin(x_1^2) + \cos(x_2^2)$$

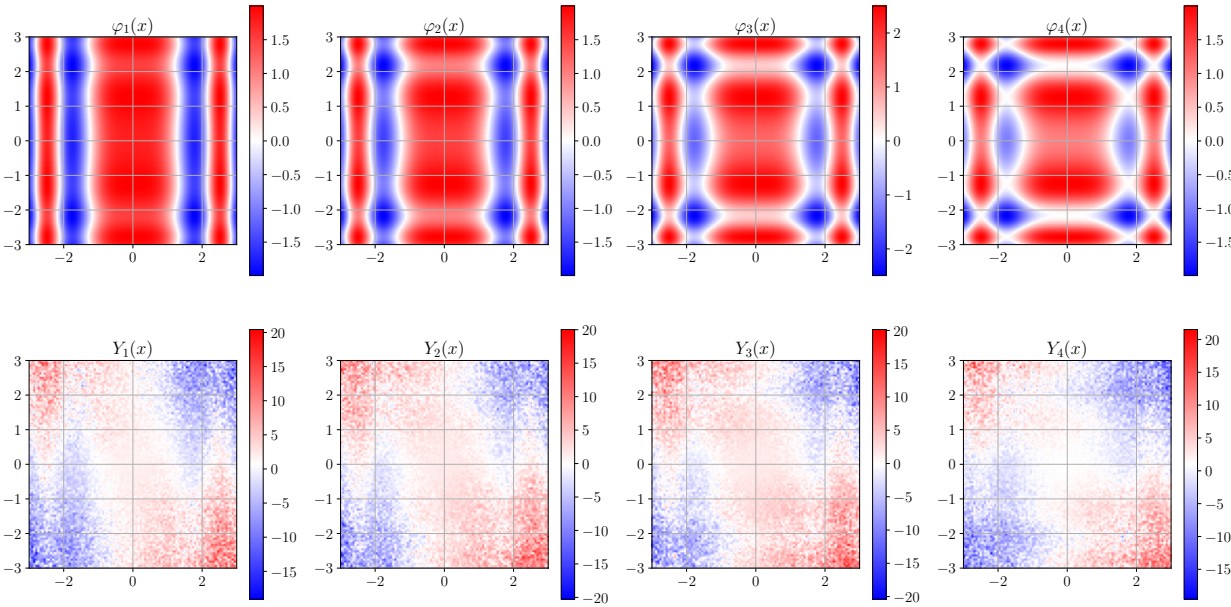

Figure 20: Reference data and observed data. Each figure represents a possible fair score to be approximated over time.

First, the model should produce a score function $\hat{\varphi}(x_1, x_2)$ as close as possible to $\varphi_1(x_1, x_2)$. As time passes, the notion of what constitutes a fair score may change, and the model's output must change properly, approximating now the second fair score function, $\varphi_2(x_1, x_2)$. Such time changes may continue to appear, and so the model should be able to track the other two fair scores, $\varphi_3(x_1, x_2)$ and $\varphi_4(x_1, x_2)$, which may appear in the future.

We used here the same neural network used for the 2-dimensional signals, training the model for 20000 epochs, using the Adam optimizer with $\mu = 1.10^{-5}$. We present the losses' evolution (Combined Loss, Wasserstein Distance and Supervised Error) in Figure 21.

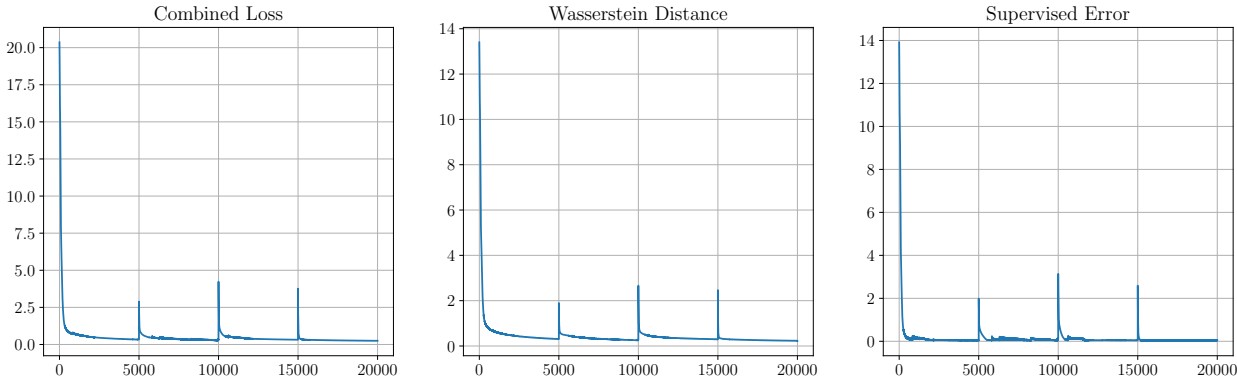

Figure 21: Losses' evolution over time.

As we can observe from Figure 21, the model tracked very well the time-changing behavior of the fair score function. We note sharp increases in the losses' values in the exact time instants where we change the fair score function (at 5000, 10000 and 15000 epochs), but as soon as the model adapts to changes, the losses' values decrease fast, indicating that the model was able to learn the new notion of fairness.

In Figure 22, we present the fair score function, the biased score and the estimated score for each one of the four functions considered. As can be observed, and in accordance with the losses' evolution already discussed, the proposed model could estimate precisely each one of the considered functions, tracking their time-changing behavior.

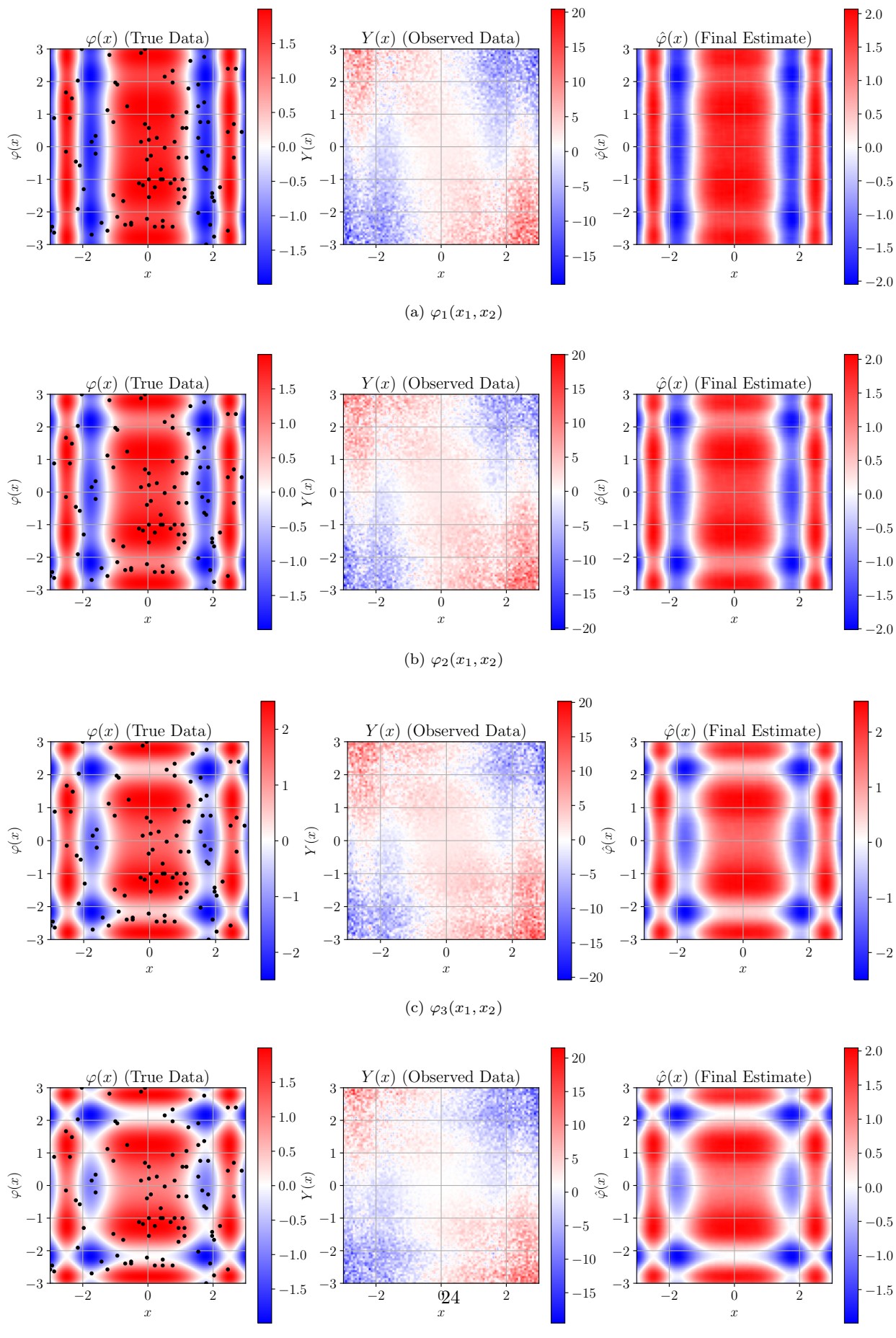

(a) $\varphi_1(x_1, x_2)$

(b) $\varphi_2(x_1, x_2)$

(c) $\varphi_3(x_1, x_2)$

(d) $\varphi_4(x_1, x_2)$

For more complex signals, the model is still able to track the changes, but it may need more epochs until the convergence to a good enough performance level. To illustrate this, let us consider a new set of four reference signals, presented below alongside their equations.

$$\varphi_1(x_1, x_2) = \sin(x_1^2) + \cos(x_2^2)$$
$$\varphi_2(x_1, x_2) = \sin(2x_1^2) + \cos(2x_2^2)$$
$$\varphi_3(x_1, x_2) = \sin(3x_1^2) + \cos(3x_2^2)$$
$$\varphi_4(x_1, x_2) = \sin(4x_1^2) + \cos(4x_2^2)$$

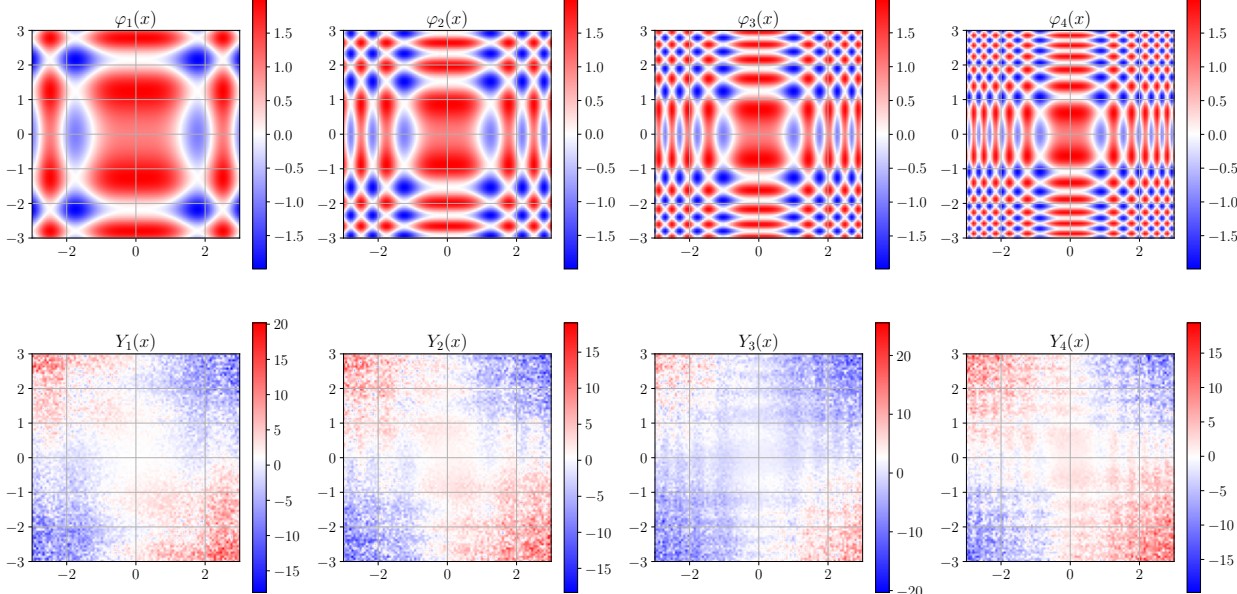

Figure 23: Reference data and observed data for more complex signals.

We repeated the same experimental setup as before, only changing the number of epochs to 40000, a number considerably higher than the previous case due to the more challenging behavior of the considered signals. As we did before, we present in Figures 24 and 25 the losses' evolution and the estimates obtained, respectively.

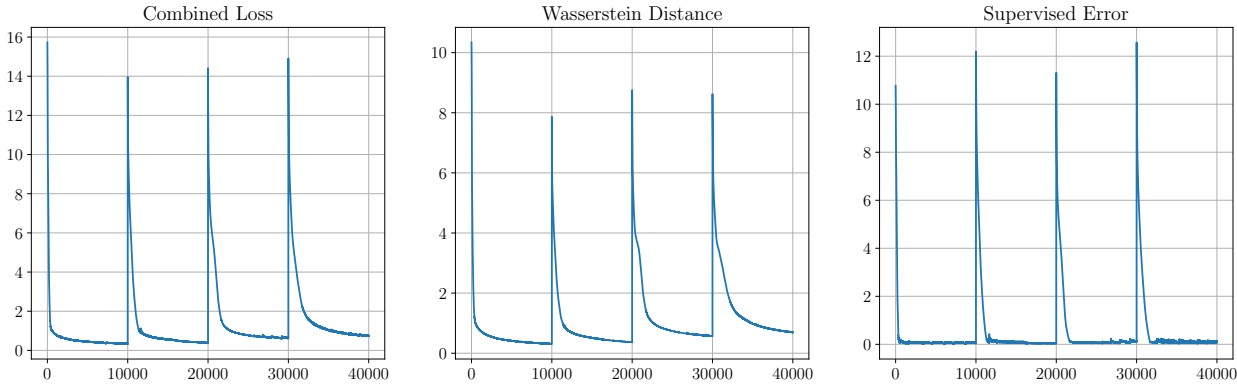

Figure 24: Losses' evolution over time for the new set of signals.

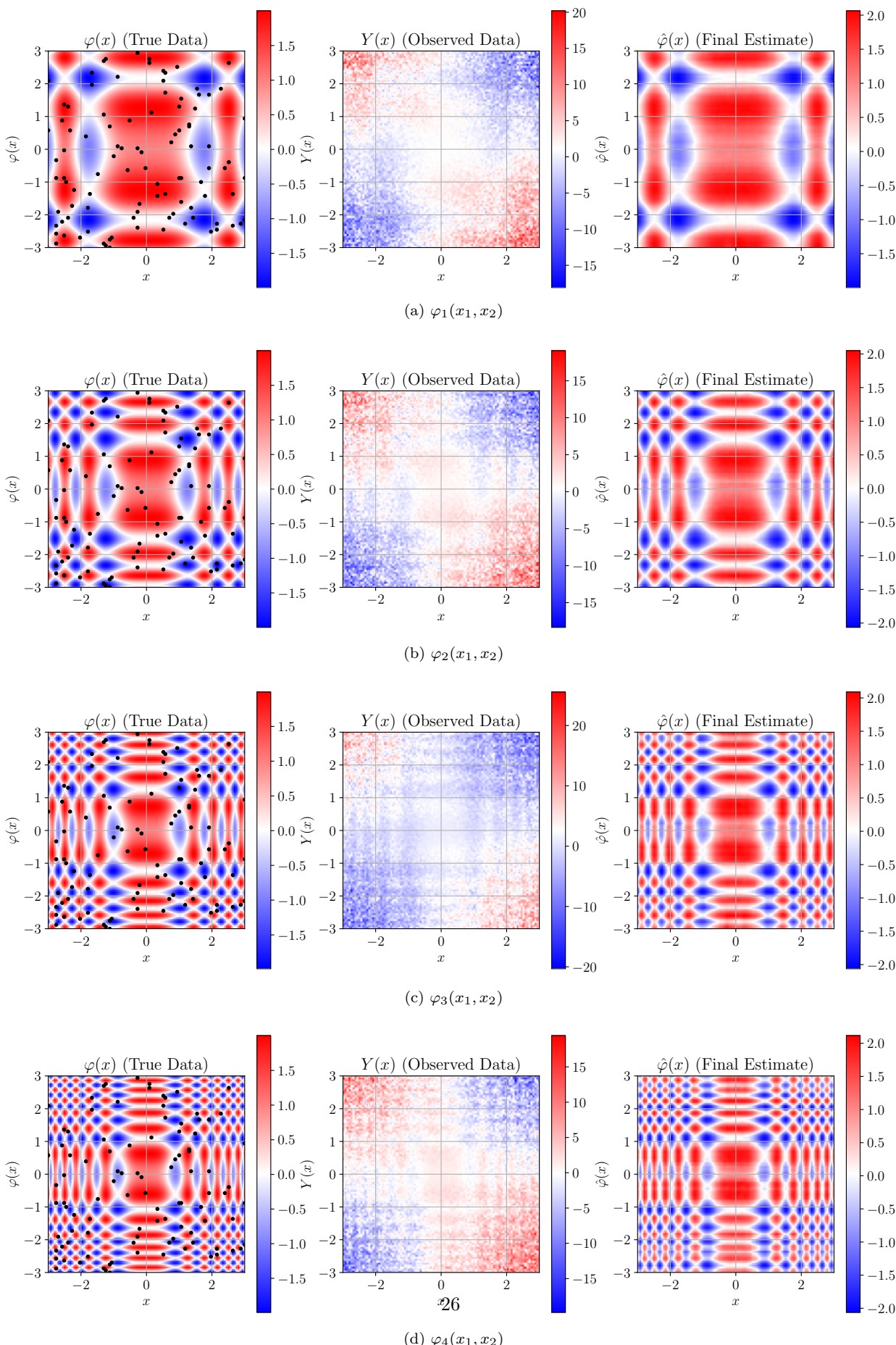

(a) $\varphi_1(x_1, x_2)$

(b) $\varphi_2(x_1, x_2)$

(c) $\varphi_3(x_1, x_2)$

26

(d) $\varphi_4(x_1, x_2)$

From the previous results, we can observe once again that the proposed method was able to track the time-changing behavior of the considered fairness notion, by only adjusting the number of epochs and without using more training pairs (*i.e.*, more supervised information).

## 5    Conclusions

In this work, we addressed the problem of debiasing supervised ML models by post-processing their outputs. Following the most recent results in Inverse Problems, we trained a neural network to learn how to automatically treat the bias. Here, we used the paradigm of weakly supervised learning, alongside a distributional constraint, given by the Wasserstein distance.

Besides the theoretical analysis made, we also evaluated our approach by means of numerical simulations. First, we considered 1-dimensional signals, which represents a more controlled scenario, less biased. In this case, we mitigate the bias by only minimizing the Wasserstein distance, *i.e.*, in an unsupervised manner. We also studied 2-dimensional signals, a scenario where the bias term was more complex, and we had to use a few labeled data points. We also considered a case of a time-varying notion of fairness, *i.e.*, a case where the unbiased score function changes over time.

Other than its technical importance to the problem, leading to very precise estimates by using a small fraction of labeled data, the weakly supervised learning is an interesting choice for social applications of ML models. First, it requires only a few training pairs, that could have been obtained after performing a polling on a small fraction of the whole population. Second, by performing such a polling, we are incorporating knowledge from specialists into the model, contributing to its accountability and explainability, both desired characteristics for ethical algorithms.

In future works, we will investigate how the proposed approach performs on real-world data. Also we are interested in investigating an optimal way to choose the labeled samples, a problem that has a very interesting connection with the active learning paradigm.

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
