# OpenReview forum: "Debiasing Machine Learning Models by Using Weakly Supervised Learning"
_TMLR — Rejected by TMLR_

### Review · Reviewer_g5YU · 2023-11-14

**Summary Of Contributions:**

The authors study the case of aiming to predict an outcome Y that is a function of observed features X and unobserved noise U in the presence of a hidden confounder S. Hidden confounding refers to X and U both being influenced by an unobserved confounder variable S. The authors refer to the hidden confounding as “endogeneity”.
The authors motivate the importance of the problem with a fair prediction task, where - as far as I understood - they assume outcome Y \in \mathds{R} and the hidden confounder to be a sensitive attribute S \in \mathds{R}. They approach the hidden confounder problem with an instrumental variable approach from the economics literature.
They propose a weakly supervised approach that learns to remove the confounding effect.
Weakly supervision refers to being able to learn from a small amount of unconfounded samples (X*, Y), where X* refers to the "true" features with the hidden confounding effect removed (obtained from expert knowledge) and a large amount of confounded samples (X, Y).
The authors evaluate their approach on synthetic data in two scenarios: a single-variable, and a two-variable case.

**Audience:**

Yes

**Broader Impact Concerns:**

No concerns.

**Claims And Evidence:**

Yes

**Requested Changes:**

I do not feel qualified to evaluate the theoretical guarantees presented in the paper. I have concerns regarding the clarify of writing, and the depth of discussing existing published work. In addition to addressing the comments provided under weaknesses, I request the following changes:

[C1] I suggest to revisit the motivational example in the methodological and experimental section to motivate the choices to put method and experimental results in the context.

[C2] I suggest to make the difference between bias and fairness stronger. This helps i) to understand the motivational example better, ii) provide a better understanding of where the proposed method is relevant beyond the fairness example. You also write sometimes “model bias” (p. 1). I would clarify this use of the term “bias”, as there are a lot of different types of biases (there are some survey papers on this).

[C3] I suggest a more thorough discussion of related work on mitigation of hidden confounders (both via IV and beyond), and causal fairness criteria for the motivational example. For example the authors write “In this scenario, measures like Fairness through Awareness Dwork et al. (2012) and Counterfactual Fairness Kusner et al. (2017) are interesting options.” (p. 2) It remains unclear to me, why these are interesting options.

[C4] I suggest to update citation. To my understanding of citation rules, citations not referred to directly should be placed in brackets and not in text.

[C5] I suggest to update the notation to make it more consistent.

[C6] I suggest to update the figures, such that the variables are not above the arrows but to the left and the right, indicating that the variables are not describing the relationship (arrow) but are input and/or output. I also suggest to update the figure captions to make them more self-contained. For example, add in Figure 2, what the different parameters etc. are.

[C7] I suggest to make more clear that the hidden confounder - if I understood correctly - is the sensitive attribute in the fairness scenario.


Comments on clarity:
* I am a bit confused by the term “forecast model”. The usual ML and algorithmic fairness literature I am familiar with uses “predictor” or “classifier”. Then on p. 2 the authors use “(the true label Y = 0 or Y = 1 in a binary classifier)” I would remain consistent with the name of the predictor/policy/classifier/forcasting model.
* If you introduce the short handle “ML” for Machine Learning in the first line of the paper, I would use consequently in the remainder of the paper ML instead of “Machine Learning”
* On page 1, I stumbled over “learn accurate forecasting models [… for] forecasting”. This appears a repetition to me and it’s not clear to me how a forecasting model is used to cluster.
* Introduction, p. 2: “either the same forecast or the same efficiency” - I did not understand this.
* Introduction, p. 2 “The objective of this work is to mitigate bias of forecasting models, when dealing with continuous-valued sensitive attributes.” - and also continuous-valued prediction, right?
* Introduction, p. 3, I had troubles understanding the example of cities and “non-urban” cities. I think it would be beneficial for the paper to provide one single strong example, rather than several ones. In this example, I was wondering, if you get the “fair information” from interviews, where do you get the “unfair” information from?
* Introduction, p.3, “the bias of an algorithm”, I would clarify again, which type of bias you consider as there are many different types of bias that the literature has acknowledged.
* Introduction, p.3. “Since here we are treating the mitigation of bias of Machine Learning models, we do not know, explicitly, how such a treatment is implemented.” I had difficulties to understand this sentence
* Theoretical Background, p. 5, I am not familiar with the term “unicity”
* Methodology, p. 8 “The other part of the data, represented by the gray area, was not labelled and should be used in an unsupervised manner” - I am not sure what this sentence refers to, I do not see a gray area in Fig. 3


Typo:
* p. 5, first paragraph last word “w”

**Strengths And Weaknesses:**

Strengths:

[S1] The paper motivates the approach by the need for unfairness mitigation strategies that can address problems that have both continuous target Y to be predicted as well as continuous sensitive attribute(s) S

[S2] The paper addresses the problem of the removal of hidden confounder information, which is an important problem that has been recognised by prior literature and appears to me understudied. It is also a practical problem that is very likely to occur in the real world.

[S3] I think it is also interesting to consider the case, where the sensitive attribue is unobserved and acts as a hidden confounder.

[S4] The paper treats the problem, where the decision making model is black box and cannot be altered. They thus propose a post-processing mechanism. I can follow this logic and believe that there might be many practical real world cases, where the model is black box.

[S5] The paper suggests a weakly supervised method, acknowledging that labeled (here data without the influence of the hidden confounder) is scarce and expensive to get. They therefore propose a method that learns to remove the influence of a hidden confounder from a prediction with access to only a small number of unconfounded data that might be expensive to obtain. The authors also provide an example, how this data could be obtained in their motivational example.

[S6] The authors evaluate their proposed method empirically providing two different synthetic examples that vary in difficult and clearly show the different steps of their empirical analysis.


Weaknesses:
[W1] While the paper motivates the approach with fairness, the given examples are not revisited at later stages of the paper (e.g., in the. methodological or experimental section). It appears to me that the motivational example is disconnected from the method and experimental evaluation. I believe drawing a connection from the motivational example to the experiments would be especially beneficial as the experiments are purely synthetic.

[W2] I am missing drawing a clear connection from the fairness example and scenario to the fairness concepts of the related literature, especially causal fairness criteria such as counterfactual fairness, both from a formal modeling perspective (how does the proposed unfairness/bias mitigation strategy relate to counterfactual fairness) as well as from the related work perspective (what prior work has done and what the paper does differently). There is also prior work on unobserved sensitive attributes that may be relevant for this paper. In this regard I also miss a clear distinction between bias and unfairness.

[W3] From p. 2 paragraph 6, remains unclear to me what exactly “endogeneity” is. Since this is the scenario studied, it would be beneficial to provide a clear definition of this concept. I understood that its “hidden confounders”, which is a term more familiar to me.

[W3] The notation seems to me often inconsistent. For example, 1) p. 4, $phi*$ appears in the equation, but as far as I see, it is not introduced before or explained in words what it is, 2) p. 5 Eq. (3), why is there sometimes a conditioning on $\mathbf{W}$ and sometimes on $W_1, W_2, \dots W_k$. Why is there in Eq. (4) no r(\mathbf{W}), but a line above r is introduced as $r(\mathbf{W})$?, 3) p. 6, First the distribution of Y is introduced as ${P}(Y)$, then later below  ${P}_Y$ is used. This is a non-exhaustive list. I suggest going over the notation and assuring that it’s consistent and that all terms are introduced.

[W4] The experimental section, though easy to follow (see [S6]), it is not apparent to me, why the authors report results on the training data set. Usually ML papers report results on the test dataset only.

[W5] The assumptions of the paper are sometimes not transparent. For example, the authors could make it more clear that the assumptions of the “labeled” data is that it’s drawn i.i.d. - at least to my understanding. Please correct me, if I am wrong. The paper does also not thoroughly discuss the limitations of the approach.

[W6] It is unclear to me, what would be the follow-up of this work. I would appreciate a paragraph that would highly the outlook and potential expansions of the paper.

[W6] Technical comments
1) Introduction: To my knowledge, the EU AI act is not yet implemented but proposed. I am not sure, if this gets clear from the writing. It might be beneficial to highlight.

2) Introduction: “Note that this application may have a strong impact on individual’ lives and will therefore be likely to be ranked as High risk by the A.I. act, so they will be regulated by the articles 9.7, 10.2, 10.3 and 71.3 of this act.” - I think this is a very strong claim. I am not a lawyer, but I would be careful to make such statement for a regulatory framework that - if I am correctly informed - is not yet implemented and there is practical evidence for this - or I would provide a reference that confirms this.

3) Introduction: The authors cite equalized odds (Hard et al, 2016) and equal opportunity (Verma & Rubin , 2018). To my understanding equal opportunity should be cited as (Hard et al., 2016) as well. To my knowledge, Verma & Rubin is a type of survey paper that just explains previous fairness notions.

4) Theoretical framework: 1) “mitigating the corresponding bias has become a legal constraint when the sensitive variable is a prohibited variable, such as gender, political or religious orientation, or race.” - Which legal framework currently into place prohibits political orientation? From a historical perspective, I am worried about this claim. There is no citation provided.

---

> ### Author Response · Authors · 2024-01-22
> **Answer to Reviewer g5YU - Part 1**
>
> **ANSWER TO REVIEWERS**
>
> **Transactions on Machine Learning Research (TMLR) Paper 1624**
>
> This official comment contains the responses to the queries, change requests, and comments made by the reviewers on article number 1624 submitted to TMLR. We answered each one of the reviewers, LXHF, g5YU and aeGX, in the different sections that follow.
>
> We, the authors, would like to express our gratitude for all the work done by the reviewers through their insightful comments, suggestions, and corrections, which significantly enhanced the quality of our work.
>
> Besides this official comment, we uploaded a new version of the paper in the forum page. We hope that with the implemented changes, the paper is now suitable for publication in this prestigious journal, TMLR.
>
> Best regards,
>
> The authors.
>
> \_\_\_\_\_\_\_\_\_\_\_\_\_\_\_\_\_\_\_\_\_\_\_\_\_\_\_\_\_\_\_\_\_\_\_\_\_\_\_\_\_\_\_\_\_\_\_\_\_\_\_\_\_\_\_\_\_\_\_\_\_\_\_\_\_\_\_
>
> **Reviewer g5YU**
>
> [W1] While the paper motivates the approach with fairness, the given examples are not revisited at later stages of the paper (e.g., in the. methodological or experimental section). It appears to me that the motivational example is disconnected from the method and experimental evaluation. I believe drawing a connection from the motivational example to the experiments would be especially beneficial as the experiments are purely synthetic.
>
> **A:** We would like to thank the reviewer for this suggestion. We have added a paragraph at the beginning of section 4 linking the experimental setup and the motivation example.
>
> [W2] I am missing drawing a clear connection from the fairness example and scenario to the fairness concepts of the related literature, especially causal fairness criteria such as counterfactual fairness, both from a formal modeling perspective (how does the proposed unfairness/bias mitigation strategy relate to counterfactual fairness) as well as from the related work perspective (what prior work has done and what the paper does differently). There is also prior work on unobserved sensitive attributes that may be relevant for this paper. In this regard I also miss a clear distinction between bias and unfairness.
>
> **A:** In fact, in this work, we propose a notion of fairness different from the counterfactual fairness concept. Our proposal involves incorporating external knowledge into the model, knowledge derived from a group of experts comprising sociologists, lawyers, economists, and other professionals capable of determining what is fair in a given context. We train the model in a way that the produced results align with the information provided by the external knowledge source. This approach was also undertaken in the work of Kletti et al. (2022), albeit within the context of fairness in ranking systems. To provide better clarification on this matter, we have rewritten the respective paragraph on page 2, and we hope it is now clearer.
>
> [W3] From p. 2 paragraph 6, remains unclear to me what exactly “endogeneity” is. Since this is the scenario studied, it would be beneficial to provide a clear definition of this concept. I understood that its “hidden confounders”, which is a term more familiar to me.
>
> **A:** Indeed, as it was stated, the idea of endogeneity was confusing. Now, in page 2, in the paragraph that starts with “*In such a setting…*” we present a clearer definition of endogeneity and we also provide two references that could be used for those interested in this phenomenon.
>
> [W3] The notation seems to me often inconsistent. For example, 1) p. 4,
>
> appears in the equation, but as far as I see, it is not introduced before or explained in words what it is, 2) p. 5 Eq. (3), why is there sometimes a conditioning on
>
> and sometimes on
>
> . Why is there in Eq. (4) no r(\mathbf{W}), but a line above r is introduced as ?, 3) p. 6, First the distribution of Y is introduced as
>
> , then later below
>
> is used. This is a non-exhaustive list. I suggest going over the notation and assuring that it’s consistent and that all terms are introduced.
>
> **A:** We changed the notation all over the paper and we hope that it is clearer now.
>
> [W4] The experimental section, though easy to follow (see [S6]), it is not apparent to me, why the authors report results on the training data set. Usually ML papers report results on the test dataset only.
>
> **A:** We agree that ML papers focus on accuracy on test datasets. Yet in our case when we deal with the estimation of functions, we think that showing the difference in the quality of the estimation of the function over the training and the test dataset enables to better understand the effect of the learned regularization.

---

> > ### Author Response · Authors · 2024-01-22
> > **Answer to Reviewer g5YU - Part 2**
> >
> > [W5] The assumptions of the paper are sometimes not transparent. For example, the authors could make it more clear that the assumptions of the “labeled” data is that it’s drawn i.i.d. - at least to my understanding. Please correct me, if I am wrong. The paper does also not thoroughly discuss the limitations of the approach.
> >
> > **A:** We agree with the referee that the selection of the points where we observe the full information (the fair score) could play an important role. In this paper we chose i.i.d uniformly chosen random points and we study the effect of the number of observations. This assumption is make clearer on page 17. We are convinced that an optimal choice of the points is interesting and could be achieved in an active learning setting. This will be the topic of a second work.
> >
> > [W6] It is unclear to me, what would be the follow-up of this work. I would appreciate a paragraph that would highly the outlook and potential expansions of the paper.
> >
> > **A:** In the Conclusion, we added a paragraph stating the follow-up of the work: applying the proposed method in real-world data and using the paradigm of activer learning to select the labeled data points.
> >
> > [W6] Technical comments
> >
> > Introduction: To my knowledge, the EU AI act is not yet implemented but proposed. I am not sure, if this gets clear from the writing. It might be beneficial to highlight.
> >
> > **A:** The Eu Act has been voted in December and the text made available yesterday. Some national discussions remain for the full implementation.
> >
> > Introduction: “Note that this application may have a strong impact on individual’ lives and will therefore be likely to be ranked as High risk by the A.I. act, so they will be regulated by the articles 9.7, 10.2, 10.3 and 71.3 of this act.” - I think this is a very strong claim. I am not a lawyer, but I would be careful to make such statement for a regulatory framework that - if I am correctly informed - is not yet implemented and there is practical evidence for this - or I would provide a reference that confirms this.
> >
> > Introduction: The authors cite equalized odds (Hard et al, 2016) and equal opportunity (Verma & Rubin , 2018). To my understanding equal opportunity should be cited as (Hard et al., 2016) as well. To my knowledge, Verma & Rubin is a type of survey paper that just explains previous fairness notions.
> >
> > **A:** Indeed, the concept of “equal opportunity” was presented in (Hardt et al, 2016). We corrected the citation.
> >
> > Theoretical framework: 1) “mitigating the corresponding bias has become a legal constraint when the sensitive variable is a prohibited variable, such as gender, political or religious orientation, or race.” - Which legal framework currently into place prohibits political orientation? From a historical perspective, I am worried about this claim. There is no citation provided.
> >
> > **A:** Indeed, there is no legal framework that prohibits political orientation and now we have changed this misleading claim.
> >
> > I do not feel qualified to evaluate the theoretical guarantees presented in the paper. I have concerns regarding the clarify of writing, and the depth of discussing existing published work. In addition to addressing the comments provided under weaknesses, I request the following changes:
> >
> > [C1] I suggest to revisit the motivational example in the methodological and experimental section to motivate the choices to put method and experimental results in the context.
> >
> > **A:** We revisited the motivation example, rewrote it and we hope that it is clearer now.
> >
> > [C2] I suggest to make the difference between bias and fairness stronger. This helps i) to understand the motivational example better, ii) provide a better understanding of where the proposed method is relevant beyond the fairness example. You also write sometimes “model bias” (p. 1). I would clarify this use of the term “bias”, as there are a lot of different types of biases (there are some survey papers on this).
> >
> > **A:** In the first paragraph of the introduction, we clarified the sense of “bias”. Naturally, fairness is the opposite of biased.
> >
> > [C3] I suggest a more thorough discussion of related work on mitigation of hidden confounders (both via IV and beyond), and causal fairness criteria for the motivational example. For example the authors write “In this scenario, measures like Fairness through Awareness Dwork et al. (2012) and Counterfactual Fairness Kusner et al. (2017) are interesting options.” (p. 2) It remains unclear to me, why these are interesting options.
> >
> > **A:** We now added an explanation to why these are interesting options.
> >
> > [C4] I suggest to update citation. To my understanding of citation rules, citations not referred to directly should be placed in brackets and not in text.
> >
> > **A:** We used the citation template provided by TMLR

---

> > > ### Author Response · Authors · 2024-01-22
> > > **Answer to Reviewer g5YU - Part 3 (final)**
> > >
> > > [C5] I suggest to update the notation to make it more consistent.
> > >
> > > **A:** We updated the notation throughout the entire paper to make it more consistent.
> > >
> > > [C6] I suggest to update the figures, such that the variables are not above the arrows but to the left and the right, indicating that the variables are not describing the relationship (arrow) but are input and/or output. I also suggest to update the figure captions to make them more self-contained. For example, add in Figure 2, what the different parameters etc. are.
> > >
> > > **A:** We updated the figures and the caption, as requested.
> > >
> > > [C7] I suggest to make more clear that the hidden confounder - if I understood correctly - is the sensitive attribute in the fairness scenario.
> > >
> > > **A:** We detailed more the experimental setup and we hope that now it is clear which are the sensitive attributes in each case.
> > >
> > > Comments on clarity:
> > >
> > > I am a bit confused by the term “forecast model”. The usual ML and algorithmic fairness literature I am familiar with uses “predictor” or “classifier”. Then on p. 2 the authors use “(the true label Y = 0 or Y = 1 in a binary classifier)” I would remain consistent with the name of the predictor/policy/classifier/forcasting model.
> > >
> > > **A:** We changed the sentence and we hope it is clearer now.
> > >
> > > If you introduce the short handle “ML” for Machine Learning in the first line of the paper, I would use consequently in the remainder of the paper ML instead of “Machine Learning”
> > >
> > > **A:** We added the short handle and used it in some parts of the text where we though it would improve the reading.
> > >
> > > On page 1, I stumbled over “learn accurate forecasting models [… for] forecasting”. This appears a repetition to me and it’s not clear to me how a forecasting model is used to cluster.
> > >
> > > **A:** We corrected the mistake. Now, the sentence is “*[...] learn accurate statistical models which are able to learn tasks, such as classification, regression, forecasting, recommendation etc., from data*”.
> > >
> > > Introduction, p. 2: “either the same forecast or the same efficiency” - I did not understand this.
> > >
> > > **A:** We corrected the sentence, and now it makes sense.
> > >
> > > Introduction, p. 2 “The objective of this work is to mitigate bias of forecasting models, when dealing with continuous-valued sensitive attributes.” - and also continuous-valued prediction, right?
> > >
> > > **A:** Yes, that is right. We corrected the sentence.
> > >
> > > Introduction, p. 3, I had troubles understanding the example of cities and “non-urban” cities. I think it would be beneficial for the paper to provide one single strong example, rather than several ones. In this example, I was wondering, if you get the “fair information” from interviews, where do you get the “unfair” information from?
> > >
> > > **A:** We changed the terminology to “small cities” and “big cities” to clarify the issue with the example. Also, the “unfair” information is observed, i.e., we know, estimate or we have observed its consequences, which motivates the treatment of such scores.
> > >
> > > Introduction, p.3, “the bias of an algorithm”, I would clarify again, which type of bias you consider as there are many different types of bias that the literature has acknowledged.
> > >
> > > **A:** We clarified again the notion of bias, as requested. Now the sentence is “*Our focus here is to mitigate the bias of an algorithm (\textit{i.e.} the automated systematically discriminatory treatment among population subgroups) by post-processing its outputs.*”
> > >
> > > Introduction, p.3. “Since here we are treating the mitigation of bias of Machine Learning models, we do not know, explicitly, how such a treatment is implemented.” I had difficulties to understand this sentence
> > >
> > > **A:** We rewrote the sentence and we hope that it is clearer now. *“Since here we are treating the mitigation of bias of supervised Machine Learning models, we do not know, explicitly, how the ML model assigned the observed scores.”*
> > >
> > > Theoretical Background, p. 5, I am not familiar with the term “unicity”
> > >
> > > **A:** It is a synonym of uniqueness and there is no special difference in meaning between them. Since the term “uniqueness” is more frequently used, we changed “unicity” to “uniqueness” in the paper.
> > >
> > > Methodology, p. 8 “The other part of the data, represented by the gray area, was not labelled and should be used in an unsupervised manner” - I am not sure what this sentence refers to, I do not see a gray area in Fig. 3
> > >
> > > **A:** We added a new figure (it was our mistake it was not in the first version of the paper) and now the sentence makes sense.
> > >
> > > Typo:
> > >
> > > p. 5, first paragraph last word “w” **A:** We corrected the typo

---

### Review · Reviewer_aeGX · 2023-11-16

**Summary Of Contributions:**

Paper deals with the problem of mitigating the bias of model when the sensitive attribute is continuous.
The paper assumes that
- a small amount of non-biases data is available
- some property of the unbiased score and biases term

**Audience:**

Yes

**Broader Impact Concerns:**

No concerns.

**Claims And Evidence:**

Yes

**Requested Changes:**

Improve the state of the art
Properly comment the assumptions
Introduce strong baselines in the experiments

**Strengths And Weaknesses:**

Paper surely deals with and interesting topic but many issues should be addressed before considering it for publication
- state of the art is largely incomplete (e.g., [1] which deals with continuous sensitive attributes) since many key works in the literature are not even mentioned. There is a huge amount of work on this topic which should be properly referenced
- assuming the availability of unbiased data is quite a strong assumption especially in the domain of fairness where "unbiased data" is a concept that depends of time (what is fay today many not be fair tomorrow) and in any case a minimum level of bias is always there
also the assumption on the unbiased score and biases term are quite strong and and, in any case, not properly commented
- experimental results are limited and unconvincing: strong baselines are missing and large experimental analysis on real data should be performed and reported.
[1] Oneto, L. and Donini, M. and Pontil, M., IEEE International Joint Conference on Neural Networks (IJCNN), General Fair Empirical Risk Minimization, 2020.

---

> ### Author Response · Authors · 2024-01-22
> **Answer to Reviewer aeGX**
>
> **ANSWER TO REVIEWERS**
>
> **Transactions on Machine Learning Research (TMLR) Paper 1624**
>
> This official comment contains the responses to the queries, change requests, and comments made by the reviewers on article number 1624 submitted to TMLR. We answered each one of the reviewers, LXHF, g5YU and aeGX, in the different sections that follow.
>
> We, the authors, would like to express our gratitude for all the work done by the reviewers through their insightful comments, suggestions, and corrections, which significantly enhanced the quality of our work.
>
> Besides this official comment, we uploaded a new version of the paper in the forum page. We hope that with the implemented changes, the paper is now suitable for publication in this prestigious journal, TMLR.
>
> Best regards,
>
> The authors.
>
> **Reviewer aeGX**
>
> Improve the state of the art
>
> **A.:** We added more references to the paper, improving the state of the art.
>
> Properly comment the assumptions.
>
> **A:** We revisited sections 2 and 3 and reformulated them. We hope that now the paper is properly commented on the assumptions.
>
> Introduce strong baselines in the experiments
>
> **A:** We now have more simulation assessing the model’s performance as well as the idea of time changing fairness.
> 11

---

### Review · Reviewer_LHXF · 2023-12-20

**Summary Of Contributions:**

The authors work in the area of bias mitigation in the situation where the model's output is real-valued (e.g., logistic regression) as well as the sensitive attributes are also real-valued (continuous).  Furthermore, the proposed work is applicable to existing models that are already deployed and are treated as a black-box; that is, the whole approach falls under the umbrella of post-processing mitigation strategies. The proposed method is based on Inverse Problems. Part of the mitigation idea is based on a loss function that balances a penalty term for labeled data as well as a bigger pool of unlabeled data (and hence the authors work in a semi-supervised / weakly-supervised setting).  The "mitigator" is assumed to have access to a pool of experts or the opinion of experts on certain inputs that could therefore provide, e.g., the "score" of a logistic regression model, on a few inputs, and these examples (labeled instances) would constitute the labeled portion of the dataset to be used by the "mitigator". The authors provide a theorem, in terms of a 1-Wasserstein distance, between the mitigated predictions and the true score of individual instances.  Ultimately the authors perform experiments with 1- and 2-dimensional input data.

**Audience:**

Yes

**Broader Impact Concerns:**

No major concerns that I can think of.

**Claims And Evidence:**

No

**Requested Changes:**

Page 1: what kind of accuracy do we have on clustering algorithms that are mentioned a bit earlier? Why mention clustering at all?

Page 2: where x represents the other attributes (non-sensitive ones)
- Y(x) denotes the model's output but this is probably bad notation because earlier above they have "the true label Y = 0 or Y = 1...".  Also, why not use Y(<x, s>), and include the sensitive attributes as part of the prediction?  To put it differently, I think you are missing one sentence somewhere in the text in the beginning to indicate that s is not used for the actual prediction.  Which, of course, brings us back to the original problem, as to how the mitigation strategy cares about s being (or not) continuous...

Page 3:
- Last paragraph of Section 1.  Perhaps give a label such as "Layout of the Paper." or something along these lines, so that it is easy to distinguish it from the rest of the text.
- Section 2, Figure 1.  This should correspond to a supervised learning model, rather than the more general term "machine learning" that is used in the paper.

Page 4:
- none of the examples given at the top paragraph are continuous.  Is the problem well-motivated?

Page 5:
- "Inverse Problems" are mentioned but are not really defined.  This is a major concern for the paper!

- Using $\mathcal{X}_L$  for the set of all labelled data seems bad notation, since lowercase x are the non-sensitive attributes of instances.  In fact, it is not what is even mentioned in the text since $i\in\mathcal{X}_L$ and if it is anything it has to be some set of indices as $i$ is used as a subscript.

- "for a small fraction of them" -> in the equation on display that follows we have "for all i..." => something is wrong.

- Section 2: In general, it is quite unclear what the goal of the mitigator is in Section 2.  I think the discussion that comes near the end and characterizes the semi-supervised nature of the problem should appear earlier in the section.

Overall, I think the paper needs major revision so that the ideas are presented in a better way to the audience.

Page 6:
- "We have modeled x as a continuous variable which is well suited to represent continuous values, such as age, financial status or ethnic proportions," -> In page 3, x is defined as a p-dimensional vector. Also, the examples are perhaps sensitive attributes where the authors reserve the letter s...


Page 7:
Why would the unbiased score all of a sudden will become distributed, rather than a single value as it was earlier?
- what is a "push forward operation"? You need a definition.
- 1-Wasserstein distance should be defined as well.
- "Supervised Learning" -> use lowercase everywhere; same is true for weak supervised learning, etc.

Page 8:
- How do you define a loss function? How do loss functions look like?  What kind of arguments do they take and what do they return?  Why does equation 5 even make sense?  Another missing definition.  Also, don't you want to give a name to that function? After all, you use it as a guideline for the different experiments that you perform in the experiments section.
- You cite "we employed the so called Weakly Supervised Learning Zhou (2018)" -> First of all, perhaps use lowercase for "weakly supervised learning".  But more importantly devote a paragraph somewhere and clearly explain what the framework is and perhaps what the important points or assumptions are that you want to adopt in your work. Weakly supervised learning can have different variants and it is usually not a matter of being "employed" somewhere, rather it captures the state in which learning is happening (e.g., few labeled examples, compared to large unlabeled instances). Do you imply anything about the loss function that you are using? For example, something along the paradigms of [1]?
[1] Deep Learning via Semi-Supervised Embedding. Jason Weston1, Frederic Ratle, Hossein Mobahi, and Ronan Collobert.
- "represented by the gray area" -> what gray area are you referring to?

Page 9-10:
While I did not check the proof of the theorem thoroughly, I do like the result.

Page 11:
- "Silverman’s rule-of-thumb" -> what is it? Why is it useful? Perhaps another thing to be put into a preliminaries section together with the loss function, Wasserstein distance, and other omitted definitions.

Pages 12-13:
- In Figures 4, 5, and 6, there are no explanations as to what the different colors represent in the diagrams on the right. Presumably with orange we see what the model learned, but this is not stated anywhere.
- Also, while we are at it, does the selection of the quadratic function make sense?  Y(x) is the estimated score so that subsequently one can decide based on a threshold for the label that will be assigned for the particular x.

Page 13:
- "that is not enough accurate " -> "that is not accurate enough"
- ReLU -> insead of citing a book, just define ReLU earlier in the text in a section with definitions and notation that you are using in the paper. Also, at the end of page 19 you are citing another source for ReLU...

Page 14:
- In Figures 8 and 9, what do we see with the different colors in the graphs?  Also, are you using blue and orange to depict learned data / predictions compared true data? Why would you do that?

General comment about experiments:
- I understand that the experimental results that you get for a 1-dimensional and a 2-dimensional case are interesting. However, are they related to any real-world example? Also, would it not make sense for your work to include some experiments on real-world data as well?
- I do not have a strong opinion on real-world data.  However, I think a paragraph is needed at the end of each of the two experiments - or even better, at the beginning of each subsection in the experiments - detailing in a brief paragraph what the goal has been, what was observed from experiments, and what conclusions can be drawn and potentially what techniques seem to work / not work in every case (should there be any experimental choices). But this has to be brief and concise. The way the results are presented right now is just a sequence of what has been done.  Which is ok in the beginning, but still needs refinement so that your message is better conveyed to the audience.

Typos:
- Page 2: "variable bu as ..." -> "variable but as..."
"character- istics as age..." -> "... such as ..."
- Page 4: "two hypothesis" -> "two hypotheses"
- Page 5: "unrolling inverse problem" -> missing a "the"? "unrolling the inverse problem"?
- Page 8: "such sorting step can not " -> "... cannot ..."

**Strengths And Weaknesses:**

Strengths:
- Interesting concept.
- A theorem exists that provides generalization guarantees for the mitigation strategy.
- Experiments in the 1-d case seem good.

Weaknesses
- While the authors hint for regression models, nevertheless, the whole paper is revolving around the idea of a "scoring" function (with interpretation a la logistic regression) and ultimately it is used for classification.
- In the beginning the authors sell as part of the innovation the fact that the sensitive attributes s are continuous. However, it is unclear how this is taken into account in the mitigation strategy since the equations depend on x (the non-sensitive attributes).  To put it differently: what difference would it make if the s vector would consist of ordinal or categorical attributes?
- The paper is lacking a clear section on notation and definitions for frequently used terms.
- What is also missing a (small?) section on methods and approaches that are used as subroutines of the mitigation strategy employed but are not well-defined in the actual text.
- In many cases, selecting text from the pdf file does not allow "correct" part of the text to be selected (though, this is a very minor concern and the issue is apparent if one wants to highlight certain passages using acrobat reader).

---

> ### Author Response · Authors · 2024-01-22
> **Answers to Reviwer LXHF - Part 1**
>
> **ANSWER TO REVIEWERS**
>
> **Transactions on Machine Learning Research (TMLR) Paper 1624**
>
> This official comment contains the responses to the queries, change requests, and comments made by the reviewers on article number 1624 submitted to TMLR. We answered each one of the reviewers, LXHF, g5YU and aeGX, in the different sections that follow.
>
> We, the authors, would like to express our gratitude for all the work done by the reviewers through their insightful comments, suggestions, and corrections, which significantly enhanced the quality of our work.
>
> Besides this official comment, we uploaded a new version of the paper in the forum page. We hope that with the implemented changes, the paper is now suitable for publication in this prestigious journal, TMLR.
>
> Best regards,
>
> The authors.
>
> \_\_\_\_\_\_\_\_\_\_\_\_\_\_\_\_\_\_\_\_\_\_\_\_\_\_\_\_\_\_\_\_\_\_\_\_\_\_\_\_\_\_\_\_\_\_\_\_\_\_\_\_\_\_\_\_\_\_\_\_\_\_\_\_\_\_\_
>
> **Reviewer LHXF**
>
> Page 1: what kind of accuracy do we have on clustering algorithms that are mentioned a bit earlier? Why mention clustering at all?
>
> **A:** As a matter of fact talking about clustering was misleading since we consider a regression scheme.
>
> Page 2: where x represents the other attributes (non-sensitive ones)
>
> **A:** We reformulated the paragraph. In fact, since we do not use explicit knowledge about the sensitive attributes there is no need to denote some of them as x (non-senstive ones) and others as s (sensitive ones). Throughout the paper, we use x for all attributes, sensitive as well as non-sensitive ones. We have also rewritten some parts in order to avoid any kind of confusion.
>
> Page 3:
>
> Last paragraph of Section 1. Perhaps give a label such as "Layout of the Paper." or something along these lines, so that it is easy to distinguish it from the rest of the text.
>
> **A:** We inserted the suggested layout
>
> Section 2, Figure 1. This should correspond to a supervised learning model, rather than the more general term "machine learning" that is used in the paper.
>
> **A:** We removed the Figure
>
> Page 4:
>
> None of the examples given at the top paragraph are continuous. Is the problem well-motivated?
>
> **A:** We agree with the referee, we only mention now the special case where continuous sensitive variables are at stake.
>
> Page 5:
>
> "Inverse Problems" are mentioned but are not really defined. This is a major concern for the paper!
>
> **A**: We agree with the referee. In order to be self contained, we refer more explicitly to the definition of inverse problems and provide references on this. It is now (we hope) clearer that the estimation of the fair score with an endogenous noise is an inverse problem.
>
> Using for the set of all labelled data seems bad notation, since lowercase x are the non-sensitive attributes of instances. In fact, it is not what is even mentioned in the text since and if it is anything it has to be some set of indices as is used as a subscript
>
> **A:** We reformulated the notations and we now describe in better way the semi-supervised setting. \𝑚𝑎𝑡ℎ𝑐𝑎𝑙{ } ∪ \𝑚𝑎𝑡ℎ𝑐𝑎𝑙{ } now denotes the set of observations and we explain clearly where we use one set over the other.
>
> "for a small fraction of them" -> in the equation on display that follows we have "for all i..." => something is wrong.
>
> **A:** We correct this statement. Now it is clear that we have access to a small fraction of training data, only to those that belong to \mathcal{X}\_{L}.
>
> Section 2: In general, it is quite unclear what the goal of the mitigator is in Section 2. I think the discussion that comes near the end and characterizes the semi-supervised nature of the problem should appear earlier in the section.
>
> Overall, I think the paper needs major revision so that the ideas are presented in a better way to the audience.
>
> **A:** We revisited sections 2 and 3, rewrote some of their parts and we hope the paper is better presented to the audience.
>
> Page 6:
>
> "We have modeled x as a continuous variable which is well suited to represent continuous values, such as age, financial status or ethnic proportions," -> In page 3, x is defined as a p-dimensional vector. Also, the examples are perhaps sensitive attributes where the authors reserve the letter s…
>
> **A:** In the usual setting fairness is modeled using a sensitive variable S. Here we are proposing a different framework that promotes fairness without explicitly knowing which are the sensitive variables. But we assume that we observe a fair score and thus all the unfairness comes from endogenous noise which is observed together with the unbiased score. Hence it is natural to use x for all p the attributes involved in the problem and not to refer to a specific variable. The variables that create the bias are all the variables x that create the biased observation noise, referred to as U(x).

---

> > ### Author Response · Authors · 2024-01-22
> > **Answers to Reviwer LXHF - Part 2**
> >
> > Page 7:
> >
> > Why would the unbiased score all of a sudden will become distributed, rather than a single value as it was earlier?
> >
> > **A:** We wrote maybe in an unclear way. Now we state better that the unbiased score provides the definition of what should be a fair score. Hence we learn (as in a plug-in method for inverse problem) to estimate scores under the fairness constraint that the estimator is similar to the fair score. The similarity is expressed using the notion of distance between the distributions.
> >
> > what is a "push forward operation"? You need a definition. **A:** We now properly defined it.
> >
> > 1-Wasserstein distance should be defined as well. **A:** We defined it, as well.
> >
> > "Supervised Learning" -> use lowercase everywhere; same is true for weak supervised learning, etc.
> >
> > **A:** We now use lowercase everywhere.
> >
> > Page 8:
> >
> > How do you define a loss function? How do loss functions look like? What kind of arguments do they take and what do they return? Why does equation 5 even make sense? Another missing definition. Also, don't you want to give a name to that function? After all, you use it as a guideline for the different experiments that you perform in the experiments section.
> >
> > You cite "we employed the so called Weakly Supervised Learning Zhou (2018)" -> First of all, perhaps use lowercase for "weakly supervised learning". But more importantly devote a paragraph somewhere and clearly explain what the framework is and perhaps what the important points or assumptions are that you want to adopt in your work. Weakly supervised learning can have different variants and it is usually not a matter of being "employed" somewhere, rather it captures the state in which learning is happening (e.g., few labeled examples, compared to large unlabeled instances). Do you imply anything about the loss function that you are using? For example, something along the paradigms of [1]? [1] Deep Learning via Semi-Supervised Embedding. Jason Weston1, Frederic Ratle, Hossein Mobahi, and Ronan Collobert.
> >
> > **A:** We corrected the orthographic mistakes. We present the semi-supervised framework in a way where we make clear that we observe few observations from the fair (unbiased) score which yet helps us estimate a distributional constraint. This constraint (expressed using the 1-Wasserstein distance) is used as a plug-in penalty to remove the unfairness from the estimator.
> >
> > "represented by the gray area" -> what gray area are you referring to?
> >
> > **A:** We referred to a Figure that was not present in this version of the paper (it was a mistake of ours). Now, we have added the figure and term “gray area” makes sense now.
> >
> > Page 9-10: While I did not check the proof of the theorem thoroughly, I do like the result.
> >
> > **A:** We appreciate that the reviewer liked the theorem.
> >
> > Page 11:
> >
> > "Silverman’s rule-of-thumb" -> what is it? Why is it useful? Perhaps another thing to be put into a preliminaries section together with the loss function, Wasserstein distance, and other omitted definitions.
> >
> > **A:** We changed the denomination and we explain that the choice of the parameter is used by looking for zones of stability.
> >
> > Pages 12-13:
> >
> > In Figures 4, 5, and 6, there are no explanations as to what the different colors represent in the diagrams on the right. Presumably with orange we see what the model learned, but this is not stated anywhere.
> >
> > **A:** We now explained what the different colors represent: blue for the true fair score function and orange for the estimated one.
> >
> > Also, while we are at it, does the selection of the quadratic function make sense? Y(x) is the estimated score so that subsequently one can decide based on a threshold for the label that will be assigned for the particular x.
> >
> > **A:** We choose the quadratic function based on examples usually used in the Econometrics field. Our choice was based on the reference Mas-Collel et al. (1995), one of the main references of the area.
> >
> > Page 13:
> >
> > "that is not enough accurate " -> "that is not accurate enough" **A:** We corrected the sentence
> >
> > ReLU -> insead of citing a book, just define ReLU earlier in the text in a section with definitions and notation that you are using in the paper. Also, at the end of page 19 you are citing another source for ReLU…
> >
> > **A:** We now presented the definition of the ReLU function instead of citing a book. Regarding the reference in the *Remark*, we kept it since it provides not only the definition of the ReLU function, but also the properties that the neural networks that use such an activation function have.
> >
> > Page 14:
> >
> > In Figures 8 and 9, what do we see with the different colors in the graphs? Also, are you using blue and orange to depict learned data / predictions compared to true data? Why would you do that?
> >
> > **A:** We now explained what the different colors represent. Note that in the fourth plot, from left to right, it appears to be only one color, but this is not true. What we have here is that the estimate was very close to the true data.

---

> > > ### Author Response · Authors · 2024-01-22
> > > **Answer to Reviwer LXHF - Part 3 (Final)**
> > >
> > > General comment about experiments:
> > >
> > > I understand that the experimental results that you get for a 1-dimensional and a 2-dimensional case are interesting. However, are they related to any real-world example? Also, would it not make sense for your work to include some experiments on real-world data as well?
> > >
> > > **A:** In this work we proposed a new approach to deal with bias in Machine Learning. The objective, then, was to assess its main properties in a more controlled scenario, hence our choice for synthetic econometric data. However, this is not our last work in this line of research and we will explore the performance of the proposed method in real-world data in our future works.
> > >
> > > I do not have a strong opinion on real-world data. However, I think a paragraph is needed at the end of each of the two experiments - or even better, at the beginning of each subsection in the experiments - detailing in a brief paragraph what the goal has been, what was observed from experiments, and what conclusions can be drawn and potentially what techniques seem to work / not work in every case (should there be any experimental choices). But this has to be brief and concise. The way the results are presented right now is just a sequence of what has been done. Which is ok in the beginning, but still needs refinement so that your message is better conveyed to the audience.
> > >
> > > **A:** We added, at the end of each subsection, a brief and concise comment about the objectives of the experiments.
> > >
> > > Typos:
> > >
> > > Page 2: "variable bu as ..." -> "variable but as..." "character- istics as age..." -> "... such as ..."
> > >
> > > Page 4: "two hypothesis" -> "two hypotheses"
> > >
> > > Page 5: "unrolling inverse problem" -> missing a "the"? "unrolling the inverse problem"?
> > >
> > > Page 8: "such sorting step can not " -> "... cannot ..."
> > >
> > > **A:** All typos were corrected.

---

### Comment · Reviewer_aeGX · 2023-10-09

Paper deals with the problem of mitigating the bias of model when the sensitive attribute is continuous.
The paper assumes that
- a small amount of non-biases data is available
- some property of the unbiased score and biases term

Paper surely deals with and interesting topic but many issues should be addressed before considering it for publication
- state of the art is largely incomplete (e.g., [1] which deals with continuous sensitive attributes) since many key works in the literature are not even mentioned. There is a huge amount of work on this topic which should be properly referenced
- assuming the availability of unbiased data is quite a strong assumption especially in the domain of fairness where "unbiased data" is a concept that depends of time (what is fay today many not be fair tomorrow) and in any case a minimum level of bias is always there
- also the assumption on the  unbiased score and biases term are quite strong and and, in any case, not properly commented
- experimental results are limited and unconvincing: strong baselines are missing and large experimental analysis on real data should be performed and reported.

[1] Oneto, L. and Donini, M. and Pontil, M., IEEE International Joint Conference on Neural Networks (IJCNN), General Fair Empirical Risk Minimization, 2020.

---

### Decision · Action_Editor_wET5 · 2024-02-13

**Recommendation:** Reject

**Comment:**

As discussed in the "Claims And Evidence" section, there is little evidence in the submission for why this method improves over previous approaches to debiased/fair machine learning. The connection between framing and theory is weak and the empirical evaluation is very preliminary. Additionally, reviewers have raised many writing and layout issues, some of which were addressed in the revision, but many remain. In particular, the literature on fair/biased machine learning is rich and the work should be better contextualized in that literature, for example, by covering different notions of fairness and bias, and how they relate to the one studied in this paper. E.g., [1] deals with continuous sensitive attributes. Moreover, the submission contains **many** similar result plots and a lot of white space around equations and figures. A revision should aim to highlight key takeaways of the empirical evaluation, probing different aspects of the proposed approach and its relation to methods from the literature.

[1] Oneto, L. and Donini, M. and Pontil, M., IEEE International Joint Conference on Neural Networks (IJCNN), General Fair Empirical Risk Minimization, 2020.

**Audience:**

There is a TMLR audience dedicated to fair/debiased machine learning, but this submission is too premature to be of interest.

**Claims And Evidence:**

In the paper outline, the authors claim to i) analyze the performance of the approach theoretically, and ii) evaluate it numerically. It is not clear from the submission how their main result (Theorem 1) helps solve this problem since it only shows that minimizing one of the two terms in their objective is sufficient—but both terms rely on having unbiased annotations. The empirical evaluation studies only simple simulation environments and the presented results show several near-identical settings (Figures 4–8). There is no comparison with other methods, even though there are many approaches to fair machine learning in the literature.

**Resubmission Of Major Revision:**

The authors may consider submitting a major revision at a later time.